# CoastalWQL: An Open-Source Tool for Drone-Based Mapping of Coastal Turbidity Using Push Broom Hyperspectral Imagery

Hui Ying Pak [1,2], Hieu Trung Kieu [1], Weisi Lin [3], Eugene Khoo [4] and Adrian Wing-Keung Law [1,5,*]

1 Environmental Process Modelling Centre, Nanyang Environment and Water Research Institute, Nanyang Technological University, 1 CleanTech Loop, Singapore 637141, Singapore; pakh0001@e.ntu.edu.sg (H.Y.P.); trunghieu.kieu@ntu.edu.sg (H.T.K.)

2 Interdisciplinary Graduate Programme, Graduate College, Nanyang Technological University, 61 Nanyang Drive, Singapore 637335, Singapore

3 School of Computer Science and Engineering, Nanyang Technological University, 50 Nanyang Avenue, Singapore 639798, Singapore; wslin@ntu.edu.sg

4 Engineering and Project Management Division, Maritime and Port Authority of Singapore, Singapore 119963, Singapore

5 School of Civil and Environmental Engineering, Nanyang Technological University, 50 Nanyang Avenue, Singapore 639798, Singapore

* Correspondence: cwklaw@ntu.edu.sg

**Abstract:** Uncrewed-Aerial Vehicles (UAVs) and hyperspectral sensors are emerging as effective alternatives for monitoring water quality on-demand. However, image mosaicking for largely featureless coastal water surfaces or open seas has shown to be challenging. Another pertinent issue observed is the systematic image misalignment between adjacent flight lines due to the time delay between the UAV-borne sensor and the GNSS system. To overcome these challenges, this study introduces a workflow that entails a GPS-based image mosaicking method for push-broom hyperspectral images, together with a correction method to address the aforementioned systematic image misalignment. An open-source toolkit, CoastalWQL, was developed to facilitate the workflow, which includes essential pre-processing procedures for improving the image mosaic's quality, such as radiometric correction, de-striping, sun glint correction, and object masking classification. For validation, UAV-based push-broom hyperspectral imaging surveys were conducted to monitor coastal turbidity in Singapore, and the implementation of CoastalWQL's pre-processing workflow was evaluated at each step via turbidity retrieval. Overall, the results confirm that the image mosaicking of the push-broom hyperspectral imagery over featureless water surface using CoastalWQL with time delay correction enabled better localisation of the turbidity plume. Radiometric correction and de-striping were also found to be the most important pre-processing procedures, which improved turbidity prediction by 46.5%.

**Keywords:** remote sensing; water quality monitoring; software; image stitching; image mosaicking; pre-processing

## 1. Introduction

Remote sensing plays an increasingly important role in water quality monitoring due to its potential to monitor over a larger spatial scale compared to other traditional methods, such as in-situ sampling and fixed-location samplers. In recent years, Uncrewed-Aerial Vehicles (UAVs), light-weight multispectral imagers (MSIs), and slightly heavier hyperspectral imagers (HSIs) have emerged as effective alternatives to satellite imagery for monitoring water quality. UAV-borne imagery offers several advantages over satellite imagery, including the ability to conduct water quality monitoring on-demand with greater flexibility in flight schedules as well as image acquisition with higher spatial resolutions [1]. Furthermore, UAV-borne imagery can mitigate the effect of extensive cloud cover in satellite

imagery, which can hinder effective water quality monitoring, particularly in the tropical and subtropical regions, as in the case of Hong Kong [2] and Brazil [3]. Additionally, due to UAVs' flexibility in flight scheduling, they are well-suited for the monitoring of dynamic and transient events. For instance, [4,5] conducted UAV-based monitoring of turbidity plumes associated with marine dredging activities and reported that turbidity attributed to dredging activities varied spatially (range of turbidity concentration exceeded 50 FNU within 100 m) and temporally ([5] reported a change in 2% of turbidity concentration per minute), which makes satellite imagery unsuitable for turbidity monitoring under such scenarios. In contrast, the monitoring of seasonal changes in turbidity using Sentinel-2's satellite imagery [6] is adequate for environments with relatively homogenous and stable turbidity concentrations.

However, significant challenges remain in UAV-based water quality monitoring, particularly in producing orthomosaics over water bodies. Currently, most of the available software in the market utilises the presence of distinctive features in the scene for coregistration (the matching of features) based on the identification of correspondence in adjacent image frames through algorithms such as scale-invariant feature transform (SIFT), speed up robust feature (SURF), and structure from motion (SfM) photogrammetry techniques (used by popular software such as PIX4D Mapper and Agisoft Metashape, e.g., [7–9]). The presence of dense distinctive features in the scene can significantly improve the localisation accuracy of UAV imagery even under GNSS denial scenarios, where [10] demonstrated a very low error rate of 0.356 m for image mosaicking using the SuperGlue deep learning neural network that conducts image extraction and image matching.

Correspondingly, the accuracy reduces drastically when there is a lack of distinctive features in the scene, such as imaging over open seas or large water bodies [11]. In addition, poor illumination conditions and shadows, as well as low contrast imagery, can also result in mosaicking failure (e.g., SfM [12,13]). This major issue was highlighted in a recent study mapping submerged plastic in a coastal environment using the Bayspec's OCI-F push-broom hyperspectral sensor [14] when limited illumination together with the dynamic sea surface and featureless water surface resulted in the failure in orthomosaicking of push-broom imagery. As such, most of the current studies involving UAV-based water quality monitoring are conducted at near-shore regions, over small lakes/reservoirs, or flown at higher altitudes to capture more textures in the scene, such as shorelines [3,14–20]. Instead of the above feature-based approach, direct georeferencing of the UAV imagery using the coordinate information and positioning data from the GPS module and Inertial Measurement Unit (IMU) have been used to facilitate orthomosaicking.

Recent studies by [21,22] adopted a direct georeferencing method for high-resolution UAV RGB/multispectral snapshot imagery, in which the MosaicSeadron from [22] achieved an error rate (standard deviation) of 2.51 m at a ground sample distance (GSD) of 0.5 m/px, as well as increased the ground coverage to 33 hectares as compared to 13.45 hectares with the SfM to mosaic snapshot images of water bodies. However, the direct-georeferencing method for snapshot imagery may not always be applicable to push-broom imagery where the image output, metadata, and the temporal resolution of the captures and measurement of flight parameters are vastly different from that of snapshot imagery due to the nature of the imaging principles that differ significantly between push-broom and snapshot imaging (see Appendix A) [23,24]. As such, specialised software from the manufacturers of push-broom sensors are typically used instead to process push-broom imagery [14,24]. Further, the accuracy of direct-georeferencing depends highly on the accuracy of the GPS module providing the geographical coordinates [25], and any missynchronisation between the GPS module and the imager can lead to significant misalignment, as demonstrated by [26].

In light of this, this study aims to (1) establish a mosaicking workflow independent of distinctive features for push-broom imagery acquired over open seas, (2) propose an efficient method for correcting the image misalignment attributed to the time delay between the on-board GNSS and the imager, (3) evaluate the performance of each data processing step for a highly modular hyperspectral push-broom system, and (4) validate the applicabil-

ity of the developed workflow developed in this study on the retrieval of coastal turbidity in the coastal region of Singapore. An open-source code with a graphical user interface, CoastalWQL (https://github.com/pakhuiying/CoastalWQL) (accessed on 13 February 2024), is also provided for end-to-end processing and analysis for public access.

## 2. Materials and Methods

### 2.1. Study Site

Successful field surveys were conducted at the Southwestern tip of Singapore, as shown in Figure 1, where active land reclamation is currently being conducted. The land reclamation activities involve the dumping of sediments by split hopper barges and dredging of the seabed, which generates high-turbidity sediment plumes [27,28]. This provides a suitable scenario for monitoring low to high turbidity concentrations. The UAV flights were conducted in the inner basin while land reclamation activities were still active, and the water depth was less than 20 m, with generally calmer waters due to surrounding caissons separating the inner basin from the open channel of the Singapore Straits.

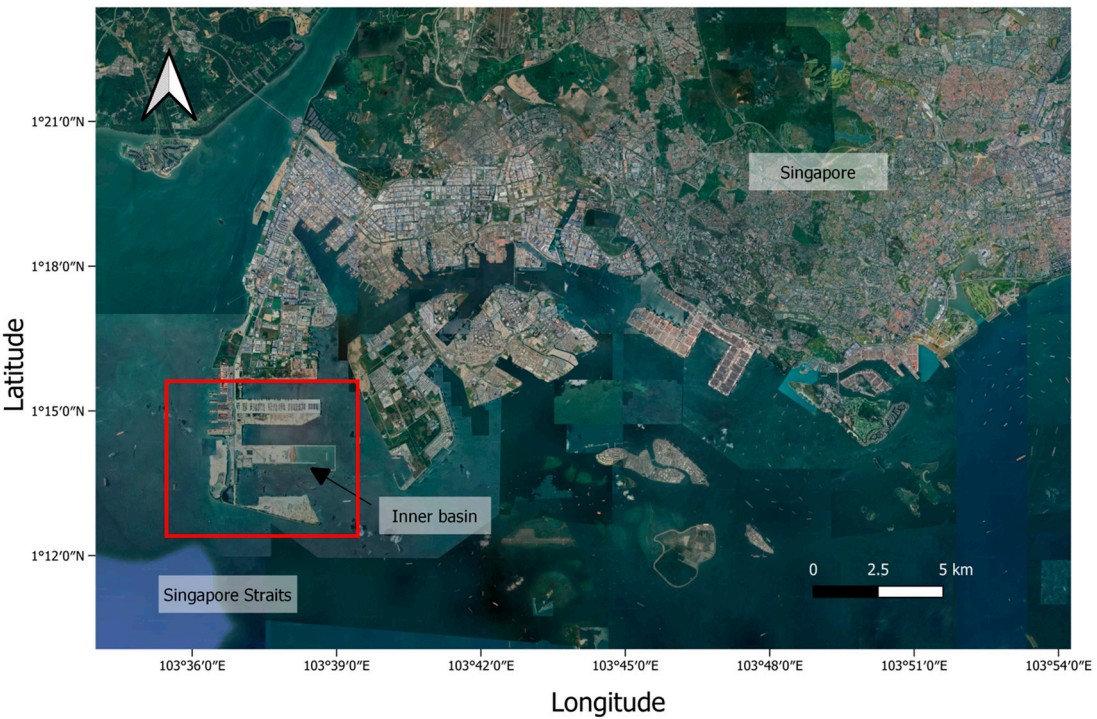

**Figure 1.** Study site at the southwest region of Singapore (enclosed by the red polygon) (Map data ©2022 Google).

### 2.2. UAV Flight Surveys and Data Acquisition

The UAV flights were conducted during the morning and the afternoon, following the logistical framework outlined by [28]. The flight's course angles were planned such that the flight course was perpendicular to the prevailing solar azimuth to avoid sun glint. The flight altitude was set at 58–60 m above ground level as the maximum flight height was imposed at 200 feet (60.96 m) by the Civil Aviation Authority of Singapore (CAAS). As such, only an area of $150 \times 150$ m$^2$ can be covered by the UAV, limited by the 15–20 min of battery life on the UAV. Parallel flight swaths (i.e., "lawn-mowing" pattern) and a ground speed of 5 m/s were configured to achieve at least 80% frontal overlap and 30% side overlap. To locate high turbidity plumes generated by transiting split hopper barges dumping sediments, a sampling vessel with a turbidity probe—YSI ProDSS (YSI, Yellow Springs, USA) was first used to detect and measure high turbidity concentrations, and the coordinates of the turbidity plume were then sent to the UAV pilot to plan a flight

survey over the turbidity plume. After the UAV flight commenced, the sampling vessel trailed behind the UAV and measured turbidity readings in the Formazin Nephelometric Unit (FNU) at one-second intervals for ground truth.

*2.3. UAV Hyperspectral System*

A rotary-wing DJI Matric M600 Pro (M600 Matrice Pro, DJI, Shenzhen, China) was deployed during the field surveys with the following payloads (Figure 2).

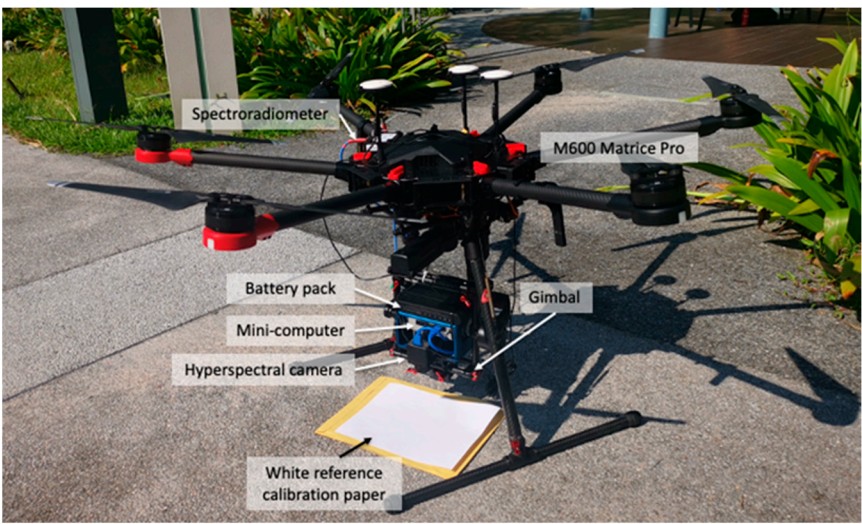

**Figure 2.** UAV set-up with an onboard hyperspectral sensor and external modules.

- A spectroradiometer (Flame VIS-NIR, Ocean Insight, Orlando, FL, USA) that covers the spectral range of 350 nm to 1000 nm was used for measuring the downwelling irradiance at one-second intervals.
- A GPS receiver module with a USB interface (U-blox 8 engine, U-blox, Thalwil, Switzerland) was used to record the geographic coordinates and the corresponding timestamps at every one-second interval.
- A commercial push-broom hyperspectral imager (HSI) (OCI-F$^{TM}$, Bayspec, San Jose, CA, USA) that covers the spectral range of 400 nm to 1000 nm (visible to near-infrared) with 61 bands was used for imaging the scene. Its field of view is 19.3° (16 mm lens) with a sensor resolution of 1024× scan-length, and its spectral resolution (Full width at half maximum (FWHM)) is 5–7 nm.
- An onboard mini-computer (Intel NUC, Intel, Santa Clara, CA, USA) was connected to the spectroradiometer, GPS module, and push-broom HIS via USB. The mini-computer was used to calibrate the spectrometer using the OceanView software and calibrate the sensor using BaySpec's SpecGrabber software in the field.
- A stabilising gimbal (Ronin MX gimbal, DJI, Shenzhen, China) to reduce any distortions to the image.

The OCI-F push-broom hyperspectral imager is a highly modular system where the spectroradiometer and GPS module data are measured independently and are not integrated with the captured images. Prior to the flight survey, sensor calibration was conducted by using a calibrated white reflectance reference in the OCI-F wavelength range (95% Lambertian reflectivity is measured against NIST traceable white reference targets) provided by the original equipment manufacturer (OEM) and a dark reference was captured with the lens cap cover.

While land reclamation activities were being carried out at the survey site, the movement of the barges within the study area made the placement of ground control points (GCP) (such as buoys in the water body) operationally unfeasible due to safety concerns. Furthermore, the UAV flight survey usually took place at a significant distance from shore as

they involved the mapping of high turbidity plumes from barge dumping operations. This frequent lack of land features often resulted in the failure of mosaicking the images [11,29] to generate a georeferenced orthomosaic using BaySpec's CubeCreator and CubeStitcher software. Therefore, an image mosaicking algorithm independent of the scene's texture is developed for the mosaicking of OCI-F's images, as described in the following sections.

### 2.4. Image Mosaicking of Push-Broom Hyperspectral Imagery

A processing workflow is created to generate the mosaic directly from the raw binary image file (Figure 3). In this workflow, each hyperspectral image is stored in a band interleaved by line (BIL) format, where the hyperspectral data of each band is stored along the columns of the hyperspectral matrix, with each hyperspectral matrix having a dimension of $1024 \times 1280$. Given 61 bands in total spread out along the columns of the hyperspectral matrix, each image band thus has a dimension of $1024 \times 20$ pixels. An image is then reconstructed by stitching the image band along the swaths (i.e., flight path) with an overlap ratio calculated from the UAV's speed and frames-per-seconds (FPS) of the sensor as detailed in the equations in Equation (1) and Table 1. The procedure is repeated for all bands to obtain a hyperspectral cube.

$$D = 2\arcsin\left(\sqrt{sin^2\left(\frac{\phi_2 - \phi_1}{2}\right) + \left(1 - sin^2\left(\frac{\phi_2 - \phi_1}{2}\right) - sin^2\left(\frac{\phi_2 + \phi_1}{2}\right)\right) \cdot sin^2\left(\frac{\gamma_2 - \gamma_1}{2}\right)}\right) \quad (1)$$

where $\phi_1$ and $\phi_2$ are the latitudes, and $\gamma_1$ and $\gamma_2$ are the longitudes of any two GPS coordinates along a swath of the flight path, which also corresponds to the central geographic coordinates of the image. $D$ represents the ground distance between the two GPS coordinates. The timestamps of the two coordinates are used to calculate the time difference between the two coordinates ($\Delta t$), and the average speed of the UAV ($v = D/\Delta t$) is calculated, which is ultimately used to determine the overall average overlap ratio between adjacent image bands along the swath (Table 1). It is noted that the UAV's speed may not be constant along the swath due to the presence of fluctuating wind conditions at the site, but obtaining the average of these variables reduced additional geometric distortions associated with random fluctuations in the UAV speed, height and FPS, which was similarly conducted by [30].

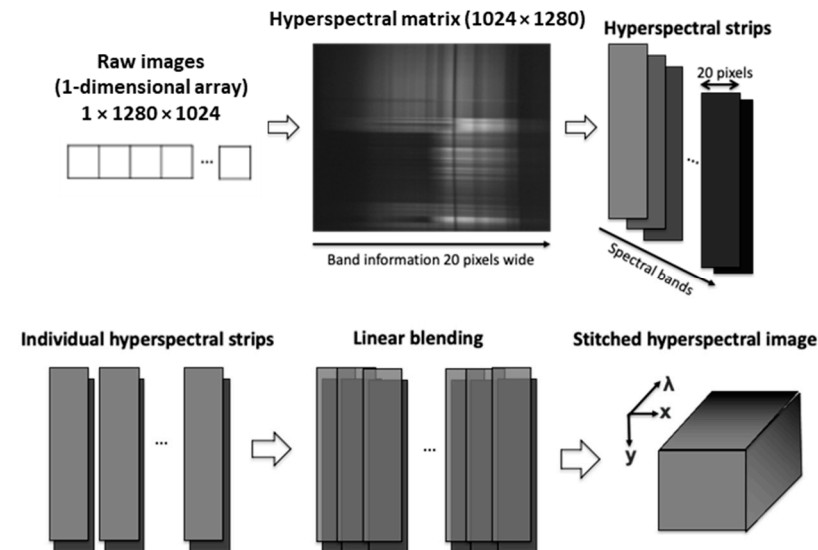

**Figure 3.** Reconstruction of the raw hyperspectral images and their mosaicking procedures.

**Table 1.** Attributes of OCI-F BaysI's HSI and calculation of overlap ratio.

| Attributes | Symbol/Dimensions/Equation |
|---|---|
| Sensor height (m) | $H$ |
| Frame rate (frames per second) | $r$ |
| Pixel size at sensor ($p$) | 5.3 μm |
| Total number of pixels in x-direction ($p_x$) | 20 |
| Total number of pixels in x-direction ($p_y$) | 1024 |
| Focal length (mm) ($f$) | 16 mm |
| Number of lines per band ($l$) | 8 |
| Actual size of sensor along x direction (mm) ($s_x$) | $(p \times p_x)/1000$ |
| Actual size of sensor along y direction (mm) ($s_y$) | $(p \times p_y)/1000$ |
| Field-of-view in the x-direction (°) ($fov_x$) | $2 \times \arctan(s_x/2f) \times 180/\pi$ |
| Field-of-view in the y-direction (°) ($fov_y$) | $2 \times \arctan(s_y/2f) \times 180/\pi$ |
| Total ground coverage in the x-direction (m) ($g_x$) | $2 \times H \times tan(\pi \times fov_x/360)$ |
| Total ground coverage in the y-direction (m) ($g_y$) | $2 \times H \times tan(\pi \times fov_y/360)$ |
| Ground resolution (m) | $g_x/p_x$ |
| Overlap ratio in the x-direction | $\frac{g_x - v/fps}{g_x}$ |

A tool is available in CoastalWQL for the user to select any two GPS coordinates along the swath and to facilitate the mosaicking of the images between the selected two GPS coordinates (Appendix B). The images located between the pair of GPS coordinates are retrieved based on matching the timestamps of the GPS coordinates and the images since every GPS coordinate and image have their corresponding timestamps. It is recommended to avoid the edge of the flight line to avoid distortions to the images, which was also suggested by [31] for mosaicking push-broom imagery. Thereafter, direct georeferencing was conducted on the image based on the calculated coordinates on each corner of the mosaicked image, where the open-source Python code can be found in CoastalWQL (https://github.com/pakhuiying/CoastalWQL) (accessed on 13 February 2024).

*2.5. Image Alignment Correction*

As mentioned previously, the images located between the pair of GPS coordinates were retrieved based on matching the images' timestamps that fall within the timestamps of the pair of selected GPS coordinates. The retrieved images were used to produce the image mosaic for each swath. However, if there is any time delay between the GNSS system and the imaging system, there would be a mismatch in the timestamps of the image captured at the location and the corresponding GPS coordinates. The images that matched the misaligned time stamps of the GPS coordinates would result in the incorrect retrieval of the set of images (Figure 4) and result in a systematic offset in all mosaicked images.

In a modular hyperspectral system where the GPS module operates independently of the hyperspectral imager, the absence of the Pulse Per Second (PPS) signal for communicating time messages between the GPS module and the imager results in poor time synchronisation between them. Therefore, there is an unknown latency associated with the time delay between creating, sending, receiving, and processing the time message from the GPS module to the imager. This could result in either a time delay in image capture or a time delay in GPS retrieval (Figure 4). While the GPS module logs the time obtained from GPS satellites, there is also an additional time delay attributed to the "clock drift" on the onboard mini-computer, which determines the timestamp of the captured image. Together, these time delays contribute to the unknown time delay offset $\Delta t$. In other words, it is challenging to determine the absolute time when the image is being captured relative to the timestamp recorded by the GPS module because the timestamps on the images do not necessarily correspond to the actual time of the image capture (Figure 4). As such, a time delay correction is needed to correct the timestamps on the images instead of using the GPS coordinates' timestamps directly. In addition, there will be more uncertainties in correcting the GPS coordinates since the time gap between adjacent GPS points (around 1 s) is significantly larger than the time gap between adjacent images (around 20 ms).

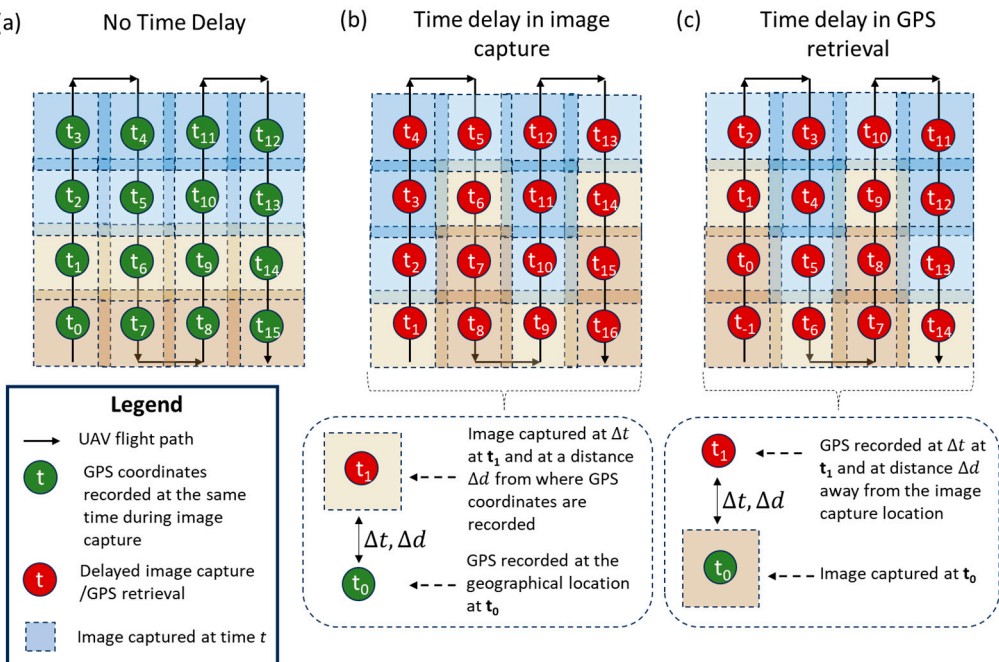

**Figure 4.** Schematic diagram of the influence of time delay on the accuracy of the image location (**a**) the ideal scenario where the time delay between image capture and GPS retrieval is absent (**b**) image misalignment caused by a time delay in image capture (**c**) image misalignment caused by a time delay in GPS retrieval (Note: Each column represents the image strip along a flight swath; colour of the tiles represents the discolouration of the water due to varying levels of turbidity concentration).

To estimate the time delay offset during the image processing step, $\Delta t$ ranging between 0 and 2000 ms with a timestep of 100 ms was iteratively added to the timestamps of the images, where $\Delta t$ can be negative or positive as follows.

$$t_i' = t_i + x\delta t = t_i + \Delta t \tag{2}$$

where $t_i'$ is the corrected timestamp, $t_i$ the uncorrected timestamp, $x$ an integer step of $x \in \mathbb{Z}$ by $\delta t = 100$ ms. The upper bound of 2000 ms for $\Delta t$ was heuristically chosen based on past flight survey experience.

For each estimated time delay, a spectral similarity of the RGB bands, quantified through Pearson's correlation, was calculated between the overlapping regions of adjacent image strips to quantify the quality of image alignment. The optimal $\Delta t$, which achieves the highest average spectral similarity across the RGB bands, could then be obtained. We note that the overlapping regions between adjacent image strips are predetermined during the flight mission planning, where the side overlap ratio is set at 30%. In the overlapping region, Pearson's correlation is conducted on each band of the RGB image mosaics as follows:

$$r(\lambda) = \frac{\sum(DN_{i,0}(\lambda) - \overline{DN_0}(\lambda))(DN_{i,1}(\lambda) - \overline{DN_1}(\lambda))}{\sqrt{\sum(DN_{i,0}(\lambda) - \overline{DN_0}(\lambda))^2 \sum(DN_{i,1}(\lambda) - \overline{DN_1}(\lambda))^2}} \tag{3}$$

where $r(\lambda)$ is the pearson's correlation coefficient as a function of the band in the overlapping region of adjacent image strips, $DN_{i,0}(\lambda)$ and $DN_{i,1}(\lambda)$, the digital numbers of each pixel in the overlapping region of two adjacent image strips, respectively, $\overline{DN_0}(\lambda)$ and $\overline{DN_1}(\lambda)$ the average digital number of the overlapping region of two adjacent image strips respectively. From Equation (3), the DN of each pixel in an image strip is subtracted

by the average DN, which makes the calculation of Pearson's correlation invariant to the brightness of the image strip.

As illustrated in Figure 4, a misalignment in adjacent image strips due to a time delay would result in a low similarity, and a time delay correction would ideally correct the timestamps on the images and hence realign the image strips by retrieving the images based on their corrected timestamps. The time delay correction is iterated with different time delays until the highest spectral correlation is achieved. The process is facilitated by the alignment correction GUI of CoastalWQL, which realigns the image strips and quantifies the spectral similarity of the overlapping regions in the adjacent image strips at each estimated time delay. In this manner, an image alignment correction can be conducted in real-time to ameliorate the problem of image misalignment (Appendix C).

### 2.6. Radiometric Correction

Varying irradiance during a field survey as a result of transient cloud cover can result in different remote sensing reflectance distributions in different image strips [32]. As such, the radiometric correction should be conducted such that the reflectance image can be comparable under different illumination conditions. The calculated reflectance is a function of the exposure settings of the sensor and digital number (DN) of the white calibration ($DN_{White}$) and dark calibration ($DN_{Dark}$) references (Equation (4)).

$$R_T(\%) = \frac{DN_{raw}/exp_{raw} - DN_{Dark}/exp_{Dark}}{DN_{White}/exp_{White} - DN_{Dark}/exp_{Dark}} \times R_{T,White} \tag{4}$$

where $R_T$ represents the reflectance image, $DN_{raw}$, $DN_{White}$, and $DN_{Dark}$ are DNs of the raw images, white reference and dark reference, respectively, and $exp_{raw}$, $exp_{White}$ and $exp_{Dark}$ are exposure times of the raw images, white reference, and dark reference, respectively, $R_{T,White}$ is the reflectance of the white Lambertian reference (95%). $DN_{White}$ was captured during sensor calibration using the 95% calibrated white Lambertian reflectance reference prior to every flight survey.

To facilitate the relative radiometric correction, various white and dark calibrations were conducted under different illumination conditions over 15 independent field surveys, and the relationship between the ratio of DN to exposure time and irradiance was modelled using a third-order polynomial curve fitting. This was similarly conducted by [2], where the DN is normalised with respect to the exposure time. The calibration curves for each wavelength were recorded in a calibration file, which was used for subsequent radiometric correction. Given the prevailing irradiance measured, the calibration curve calculates the corresponding ratio of DN to exposure time for the white and dark reference, and they are used in Equation (4) to radiometrically correct each image.

### 2.7. Sun Glint Correction

During UAV-based water quality monitoring missions, sun glint contamination can be especially prevalent. Specular reflections that result in sun glint can deteriorate the image quality by reducing the signal-to-noise ratio and limiting the accuracy of extracted spectral information, rendering the data unusable [33–35]. In particular, in coastal regions, sun glint effects can be intensified by wind-roughened water surfaces [34,36].

A sun glint correction is proposed in this study, based on the modification of the Sun Glint Aware Restoration (SUGAR) algorithm [37] to correct the sun glint in the mosaicked image. Sun glint pixels exist as bright pixels amidst a darker sea/ocean background, and thus, they manifest as a high spatial intensity gradient (high-frequency information), especially in high-resolution UAV imagery, which can be detected easily with a Laplacian of Gaussian (LoG) kernel (Equation (5)).

$$\text{LoG}(i,j) = -\frac{1}{\pi\sigma^4}\left[1 - \frac{i^2+j^2}{2\sigma^2}\right]e^{-\frac{i^2+j^2}{2\sigma^2}} \tag{5}$$

where the LoG is a kernel that approximates the second derivative of a Gaussian, $\sigma$ represents the standard deviation of the gaussian, typically set at $\sigma = 1$, and $i, j$ represents the coordinates of the LoG kernel.

High-frequency information, such as sun glint pixels, are identified by convolving the LoG kernel with the reflectance image ($R_T$) to obtain $L(x, y, \lambda)$ (Equation (6)).

$$L(x, y, \lambda) = LoG(i, j) * R_T(x, y, \lambda) \tag{6}$$

where $x, y$, and $\lambda$ represent the image coordinates of $R_T$ and the image band (wavelength), respectively. Since the LoG approximates the second spatial derivative of $R_T$, the more negative $L(x, y, \lambda)$ is, the brighter the pixel, and the higher the likelihood that the pixel is a glint pixel. As such, a negative threshold $t$ can be applied to $L(x, y, \lambda)$ to obtain the potential glint pixels (Equation (7)).

$$G(x, y, \lambda) = \begin{cases} 0, & if \quad L(x, y, \lambda) \quad \geq t \\ 1, & if \quad L(x, y, \lambda) \quad < t \end{cases} \tag{7}$$

where $G(x, y, \lambda) = 0$ and $G(x, y, \lambda) = 1$ represent pixels with no glint and glint, respectively. However, if $t$ is chosen to be too high (less negative), sun glint pixels and small variations in intensity spatial gradient, such as variations in water colour, will also be corrected, which is undesirable. As such, to be more conservative, $t$ should be chosen to be low enough (more negative) to identify bright sun glint pixels. To automate the selection of $t$, data binning is conducted on $L(x, y, \lambda)$, where the number of bins is heuristically selected as 10 (equivalent to obtaining a histogram or probability distribution function (PDF) of $L(x, y, \lambda)$) (Equation (8)). $L(x, y, \lambda)$ is a unimodal distribution, with a sharp peak at $L(x, y, \lambda) = 0$, which corresponds to a smooth spatial intensity gradient (due to $R_T$ generally having more non-glint pixels than glint pixels). Thus, any values on the left side of the distribution's peak would guarantee a negative value (Equation (9)), provided that the number of bins is sufficiently small (e.g., around 10).

$$[L_i, L_{i+1}] = argmax(\{p_1, p_2, \ldots p_i \ldots, p_n\}) \tag{8}$$

$$t = L_{i-1} \tag{9}$$

In Equation (8), $\{p_1, p_2, \ldots p_i \ldots, p_n\}$ represents the set of binned $L(x, y, \lambda)$. $p_i$ corresponds to the bin with the highest frequency (i.e., the peak of the distribution), with its corresponding interval values $[L_i, L_{i+1}]$. Thus, to ensure a negative threshold, $t$ should be smaller than the interval values of the highest frequency bin (i.e., $t < L_i$), which yields the interval value of the bin located to the right of $p_i$ (Equation (9)).

The correction of the sun glint is then conducted by propagating the neighbourhood values across the sun glint pixels since water quality in a small local region is highly spatially correlated (Equation (10)). The assumption is reasonable as the typical GSD of UAV imagery is in the scale of centimeters that is comparable to the typical length scale of the capillary wave, which is responsible for most of the sun glint artefacts for high-resolution imagery [38].

$$R_T(x, y, \lambda) = \begin{cases} R_T(x, y, \lambda), & if \quad G(x, y, \lambda) \quad = 0 \\ \min_{\{5 \times 5\}} R_T(x, y, \lambda), & if \quad G(x, y, \lambda) \quad = 1 \end{cases} \tag{10}$$

where $\min_{\{5 \times 5\}} R_T(x, y, \lambda)$ represents taking the minimum value of a $5 \times 5$ neighbourhood at $R_T(x, y, \lambda)$ to obtain the background spectra. It can be observed from Equation (10) that the correction is only conducted on glint pixels, leaving non-glint pixels unaltered. This ensures that the sun glint correction is focused only on the sun glint pixels, and other features in the scene are kept unchanged, such as turbidity plume or shoreline.

### 2.8. Removal of Stripe Noises

The presence of stripe noises is one of the most common sources of noise that largely affects hyperspectral imaging. It affects the signal-to-noise ratio and generally degrades the quality of the image. Most importantly, the presence of stripe noises affects the accuracy of the extracted spectral information, leading to inaccurate observations.

Prior to every flight survey, white and dark calibrations are performed to facilitate radiometric correction. The homogeneity of the white calibration surface allows for the quantification of the intensity and spatial distribution of the stripe noises. Therefore, it is important to ensure that no shadows are cast on the calibrated white reference during sensor calibration. The stripe noises generally occurred at the same location, as in the case with Bayspec's OCI-F push broom camera. The stripe noises were most distinct at the edges of the images, and they occurred intermittently throughout the images, as demonstrated in the 'n' shape latitudinal DN profile with sharp depressions in DN (Figure 5).

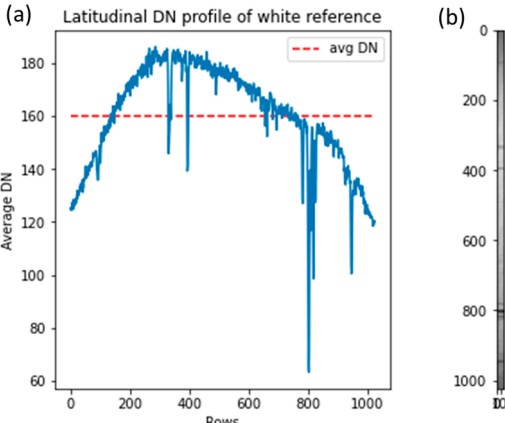

**Figure 5.** (**a**) Longitudinal average DN profile of the calibrated white reference in the mid-band where the *x*-axis represents the rows of the white reference image, (**b**) Hyperspectral push broom image strip of the white reference in the mid-band with the dimension of $1024 \times 20$.

The goal of de-striping is to flatten the latitudinal DN profile of the calibrated white reference for each band as much as possible, with respect to its maximum DN, as it is assumed that the maximum DN is unaffected by stripe noises. A de-striping factor of $\max(DN_{White}(\lambda))/DN_{White}(x, y, \lambda)$ was multiplied by each pixel in the mosaicked image for de-striping (Equation (11)). At the same time, this also provides an efficient method for correcting vignette effects at the edges of the image.

$$de - striped\ image(\lambda) = \frac{\max(DN_{White}(\lambda))}{DN_{White}(x, y, \lambda)} \times R_T(x, y, \lambda) \qquad (11)$$

### 2.9. Masking and Classification

Masking of non-water bodies is essential for UAV-based water quality monitoring, especially if in-situ sampling is carried out simultaneously. This is because vessels carrying out in-situ sampling or other vessels may be captured in the UAV imageries. Thus, extraction of spectral information from these vessels has to be avoided to prevent the introduction of noises in the extracted spectral information. Furthermore, due to the nature of the survey site in this study, vessels have to be masked for confidentiality reasons as well. The presence of high-concentration sediment plume and sun glint on the water surface may affect the result of image segmentation and hence affect the reliability of the output mask. Therefore, it is imperative that a robust and efficient image segmentation model can be used for various scenarios. An XGBoost model was trained for the classification task as follows.

A total of 66 image lines that contain vessels and/or caissons inside the scene (out of 129 image lines) were acquired over four independent field surveys on separate days with differing illumination conditions. The vessels and caissons were annotated manually by hand to produce the ground truths (labels) for the segmented classes (0 = water; 1 = vessels; 2 = land/caisson) from the 66 false-composite image lines. Therefore, 66 pairs of false composite images and their corresponding labels were produced. The XGBoost model was trained on 70% of the data and evaluated on the remaining 30% of the data.

The multi-class cross-entropy was used as the loss function for the XGBoost's evaluation metric (Equation (12)). The training of the classification model was conducted until the loss metric had not improved in 10 rounds.

$$Loss(Y, P) = -logPr(Y|P) = -\frac{1}{N} \sum_{i=0}^{N-1} \sum_{k=0}^{K-1} y_{i,k} log p_{i,k} \tag{12}$$

where $Loss(Y, P)$ represents the log loss, Y is the labels represented as 1-of-K binary indicator matrix, K represents the labels, and P is the matrix of probability estimates.

Class-wise Intersection-over-Union (IoU) (Equation (13a)), Frequency-weighted IoU (Equation (13c)), mean IoU (Equation (13d)), accuracy (Equation (13e)) and error rate (Equation (13f)) were used to evaluate the performance of the image segmentation for the above models.

$$IoU_i = \frac{TP_i}{TP_i + FP_i + FN_i} \tag{13a}$$

$$W_i = \frac{P_i}{\sum_i^C P_i} \tag{13b}$$

$$IoU_{frequency} = \sum_i^C (IoU_i * W_i) \tag{13c}$$

$$\bar{IoU} = \frac{\sum_i^C IoU_i}{C} \tag{13d}$$

$$accuracy_i = \frac{TP_i + TN_i}{TP_i + FP_i + TN_i + FN_i} \tag{13e}$$

$$Error\,rate_i = 1 - accuracy_i \tag{13f}$$

where $i$ represents the $i$th class out of $C$ total classes, $TP$, $FP$, $TN$, and $FN$ represent true positives, false positives, true negatives, and false negatives, respectively, $P_i$ represents the number of pixels in the $i$th class.

### 2.10. Assessment of Pre-Processing Methods with Turbidity Retrieval

To assess the effectiveness of each pre-processing procedure, turbidity retrieval was conducted at each step according to the following sequence: (1) uncorrected reflectance (original), (2) time delay correction, (3) de-striping, (4) radiometric correction, and (5) sun glint correction. Masking of the vessels was applied prior to pre-processing for confidentiality purposes.

Prior to turbidity retrieval, spectral reflectances were extracted using CoastalWQL from a region of 40 × 40 pixels surrounding the in-situ turbidity measurements based on their GPS coordinates, which corresponds to a ground coverage of 0.8 × 0.8 m². The spectral reflectances were then averaged within the 40 × 40 pixels. Observations with negative reflectances and invalid values were removed as they were outside the dark and white calibration values, which may lead to erroneous results. Additionally, spatially overlapping sampling points where the sampling vessel traversed the same location at least once were also removed (see Appendix D). This is because the region where the sampling vessel has traversed before has already been disturbed and may cause greater dispersion in the turbid region, affecting the accuracy of the in-situ measured turbidity and the retrieval of the turbidity.

The reflectance images were converted to turbidity in FNU using Nechad et al.'s (2010) semi-analytical turbidity algorithm (Equation (14)) [39], which is a single-band semi-analytical algorithm that is largely applicable for most turbid waters [40]:

$$Turbidity[FNU] = \frac{A_\tau R_T(\lambda_{\text{Red}})}{(1 - \frac{R_T(\lambda_{\text{Red}})}{C})} \tag{14}$$

where $A_\tau$ and $C$ are wavelength dependent calibration coefficients to be optimised using the least squares method, and $R_T(\lambda_{\text{Red}})$ refers to the reflectance values in the red or NIR band. The calibration coefficients were optimized for each set of extracted reflectances to ensure that the calibrated coefficients are not biased towards any pre-processing step.

Variants of the algorithm involving its applicability on red to NIR wavelengths were reported in various studies, where [4,21] used 660 nm and 668 nm, and [41] chose 710 nm instead. [39] suggested 681 nm, which has the lowest error rate for suspended particle matter (SPM) retrieval. As an extension of Nechad et al. (2010), [39,40] suggested an NIR band at 859 nm for turbidity retrieval in moderate to high turbid waters and proposed the red band at 645 nm band for medium to low turbid waters.

An analysis was conducted to determine the quantitative effect of these bands on turbidity retrieval. The closest bands in the hyperspectral sensor that matched these bands were 641 nm, 660 nm, 678 nm, 715 nm, and 860 nm, which ranged from the red to NIR wavelength.

The coefficient of determination ($R^2$), root mean squared error (RMSE), and mean absolute percentage error (MAPE) were quantified to assess the performance of the semi-empirical turbidity algorithm.

$$RMSE = \sqrt{\frac{\sum_{i=1}^{N}(y_i - \hat{y}_i)^2}{N}} \tag{15a}$$

$$MAPE = \frac{1}{N}\sum_{i=1}^{N}\left|\frac{y_i - \hat{y}_i}{y_i}\right| \tag{15b}$$

where $N$ refers to the number of data points, and $y_i$ and $\hat{y}_i$ refer to the observed turbidity and predicted turbidity, respectively.

In summary, the entire workflow procedure to generate georeferenced mosaics from the raw binary image files, data and image processing steps such as radiometric correction, sun glint correction, de-striping and masking, and prediction of the water quality is summarised in Figure 6. Additional information regarding the inputs and outputs of the workflow executed in CoastalWQL is available in Appendices E and F.

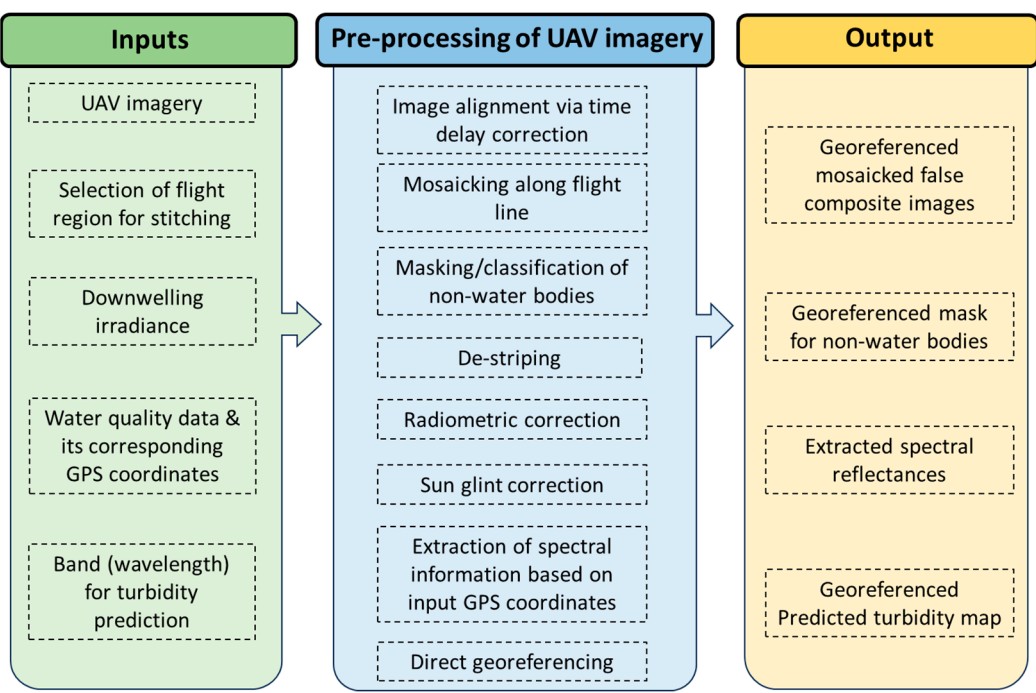

**Figure 6.** Workflow procedure to obtain reflectance products and predicted water quality (workflow executed in *CoastalWQL*).

## 3. Results

### 3.1. Evaluation of Image Pre-Processing Methods

#### 3.1.1. Image Alignment

To assess the effectiveness of the direct geo-referencing method together with the time delay correction method, hyperspectral images were taken over a scene with a diversity of features at Nanyang Lake, Singapore. The UAV hyperspectral system was flown at 35 m AGL. Distinctive features were identified as GCPs in the Google satellite imagery tiles, which serve as ground-truth images (Figure 7a). They include easily distinguishable features such as corners and edges along the wooden boardwalk and zebra crossing. The corresponding features were identified and marked in the geo-referenced UAV imagery. The alignment errors between the GCPs in the ground-truth satellite imagery and the UAV imagery without applying the time delay correction ranged from 3.15 to 8.90 m (with an average error of 5.35 m) (Figure 7b), while the alignment error with the UAV imagery after applying the time delay correction ranged from 1.07 to 8.58 m (with an average of 4.08 m, and a standard deviation of 2.69 m). The mosaicked image's central geographic coordinates were verified to match the recorded GPS coordinates by the GPS module (see Appendix G), which indicates that the alignment error was largely attributed to the lower accuracy of the longitudinal geographic coordinates recorded by the GPS module.

Figure 8 shows the effect of various time delay corrections (i.e., between the GPS timestamps and image timestamps) on the image alignment, where time delay may be a significant issue for highly modular camera systems with independent GPS and image acquisition modules. It can be observed that without any time delay correction, the systematic offset can be visibly observed in adjacent image lines in the first panels of Figure 8a,b. After applying the time delay correction, the imageries in Figure 8a,b showed the best alignment in the images after a time delay correction of 1 s, with an average spectral correlation of 0.9662 and 0.9668.

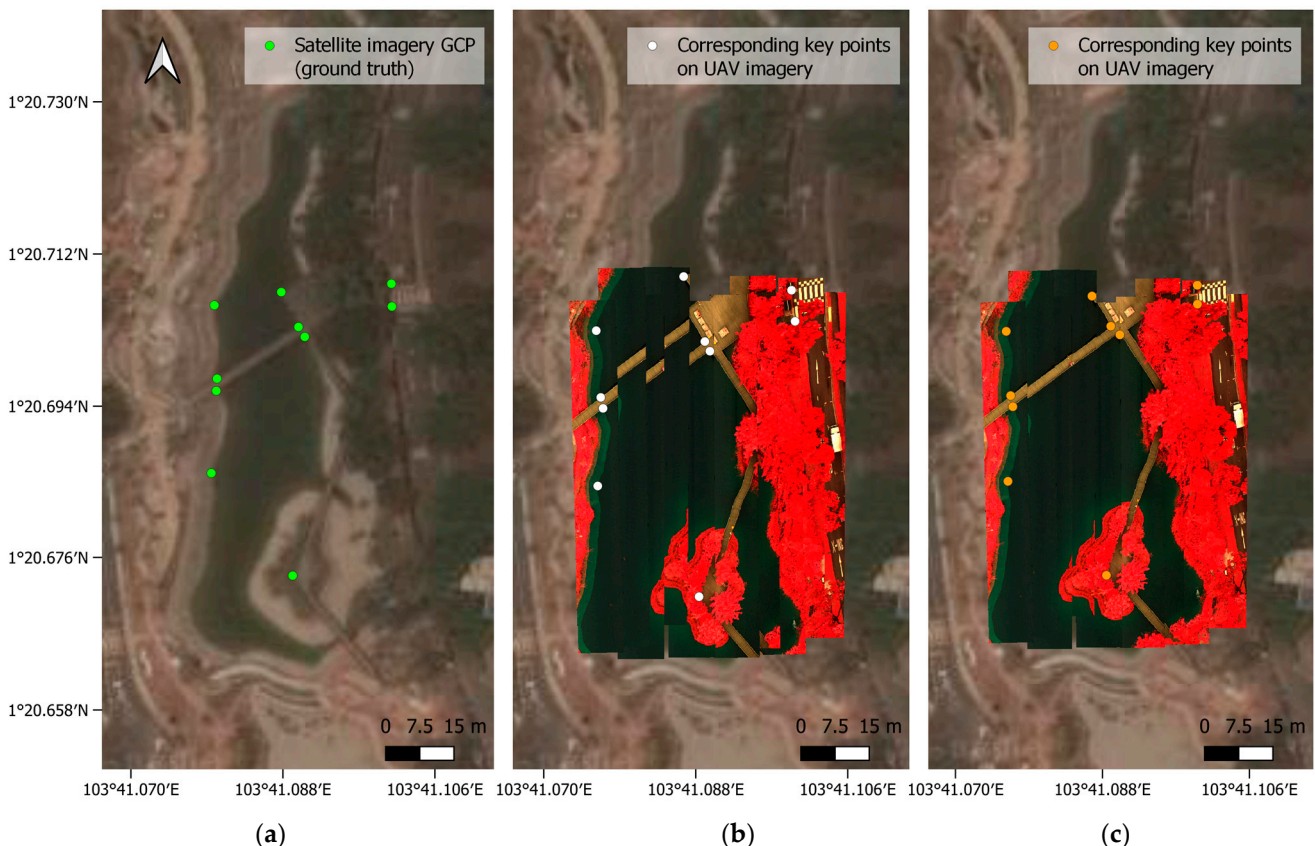

**Figure 7.** Common key distinctive features identified in Nanyang Lake, Singapore (**a**) high-resolution Google satellite imagery which serves as ground-truth (**b**) mosaicked UAV false composite image without time delay correction (**c**) mosaicked UAV false composition image with time delay correction.

Depending on the speed of the UAV, a small time delay could result in significant discrepancies between the ground truth and the imaged scene, and the misalignment attributed to the time delay may become more severe with increasing UAV speed. For instance, a time delay of one second with a UAV speed of 6 m/s could result in around 6 m of discrepancy.

### 3.1.2. Radiometric Correction—Calibration Curve

The third-order polynomial curve fitting for the white and dark reference demonstrated a strong relationship between the ratio of DN to exposure time and the measured irradiance for most of the wavelengths, with an $R^2$ exceeding 0.94 for most wavelengths (Figure 9). However, it is observed that the signal-to-noise ratio for wavelengths smaller than 450 nm and larger than 940 nm is relatively low, where the calibration curves for the white and dark reference lie close to one another, especially at low irradiance values. This also implies that radiometric correction at these wavelengths under low illumination conditions would be highly limited and could result in numerical instability, as reflectance values for the white and dark reference are very similar.

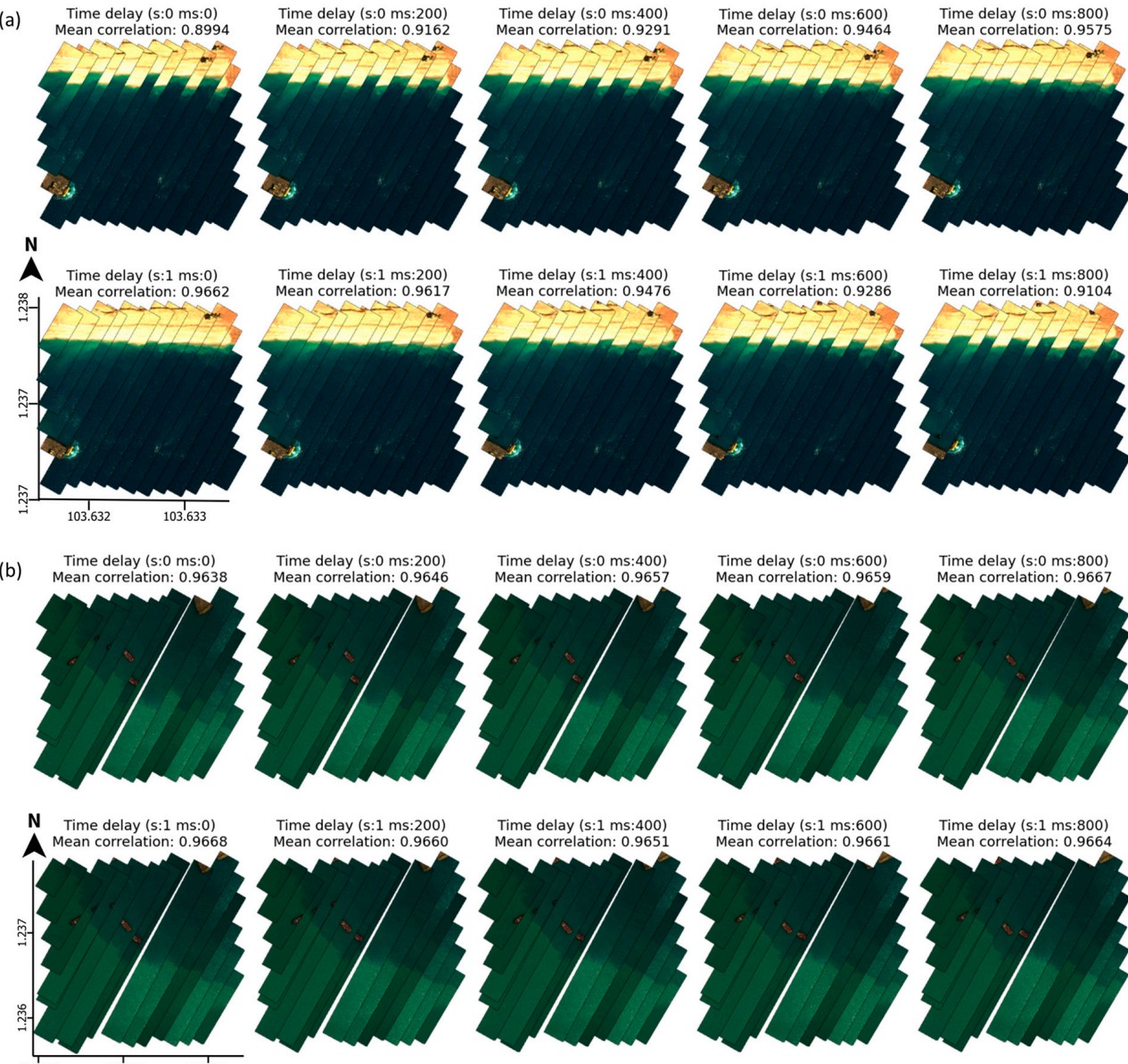

**Figure 8.** Time delay correction for flights with (**a**) visible land features, (**b**) turbidity plumes and distinctive features such as shoreline are absent in the scene (Note: the mosaicked images are raw false composite images in DN without any corrections applied).

### 3.1.3. Classification and Masking

The classification of the water body, vessels and caisson/land categories yielded 0.99 in accuracy, with the highest class error rate of 0.0087. The class IoU metric was the lowest for vessels at 0.5401, and it achieved at least 0.93 for water bodies and caisson land. The frequency-weighted IoU, mean IoU, and overall model accuracy were 0.9824, 0.6175, and 0.9903, respectively. Therefore, the trained XGBoost is deemed to be adequate for masking and classification tasks (Figure 10).

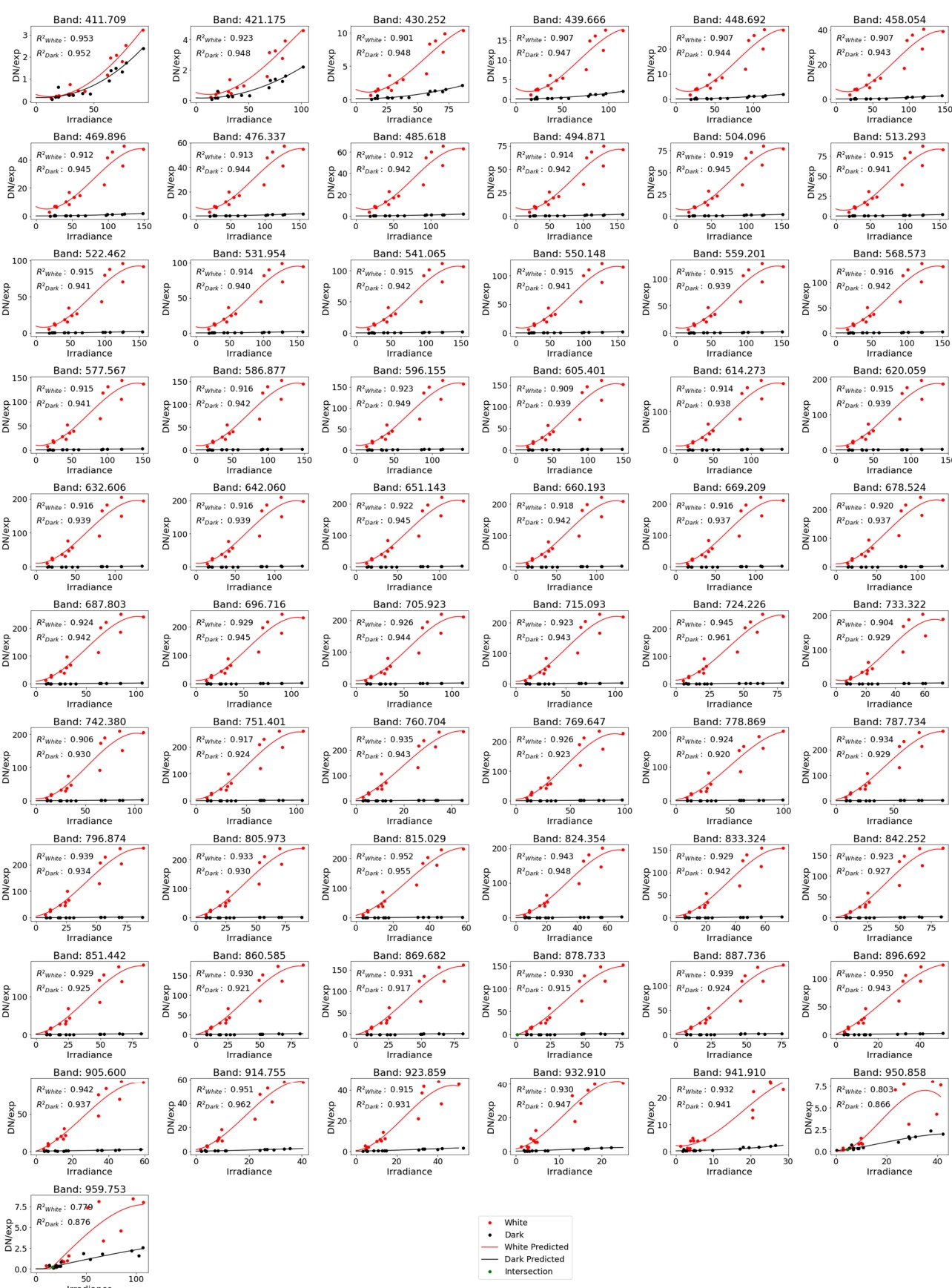

**Figure 9.** Relationship between the ratio of DN to exposure time and downwelling irradiance for each wavelength.

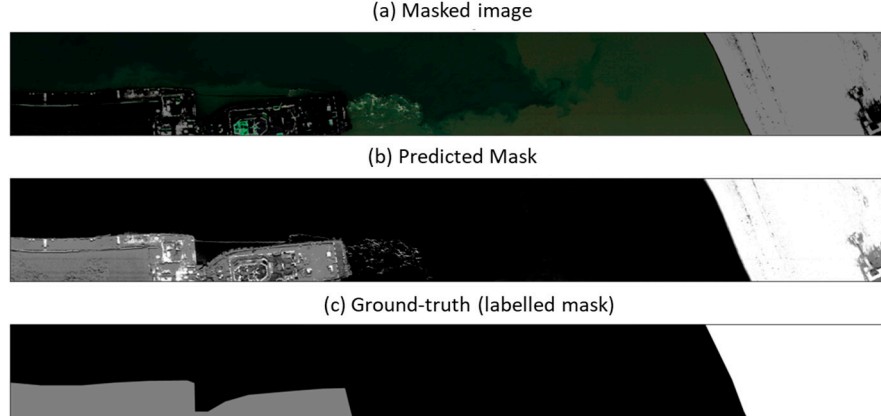

**Figure 10.** Masking and classification of non-water pixels using a trained XGBoost model.

### 3.1.4. Noise Removal—Sun Glint Correction and De-Striping

The sun glint correction employed in this study was compared with traditional NIR-based sun glint correction methods—Hedley et al.'s (2005) [42], Goodman et al.'s (2008) [43], and Kutser et al.'s (2009) [44] algorithms, which are commonly applied to high-resolution imagery. As observed from Figure 11, the NIR-based sun glint correction methods by Hedley, Goodman and Kutser significantly altered the reflectance of the turbidity adjacent to the shoreline as well as the shoreline itself. These algorithms exhibit overcorrection in these regions because the NIR-leaving radiance is not negligible in these regions, which may affect the retrieval of water quality parameters like turbidity [33]. The overcorrection is particularly prominent for Goodman's algorithm as reflectance values became negative in the extreme wavelengths, where turbidity regions near the shoreline darkened significantly. In the magnified view of the glint region, the glint pixels are still clearly visible for the NIR-based methods, but the glint pixels are much reduced for this study's method. Additionally, in the longitudinal profile of our method, the sharp peaks associated with glint pixels are significantly reduced, while sharp peaks are still present in the NIR-based methods. The total variance for all algorithms was significantly reduced, except for Kutser's algorithm, but the reduction in total variance for this study's algorithm was largely attributed to the reduction in the intensity of the glint's pixel whilst keeping the reflectance of the shoreline unaltered. As such, a texture-aware total variation-based sun glint correction such as that proposed [45] similarly overcomes the general limitation of overcorrection in turbid regions since NIR is not used as a surrogate for sun glint intensity. The sun glint correction proposed in this study is thus solely focused on the sun glint pixels while leaving other features in the scene unaltered.

It is also noteworthy to mention that modelling statistical models of the sea state, such as the Cox and Munk model [46], together with radiative transfer theory to predict the distribution of sun glint, generally performs poorly for high-resolution UAV imagery [33]. This is because the spatial distribution of the high-resolution sun glint pixels cannot be located exactly using the statistical sea state model [47]. Thus, a pixel-based correction is generally a more effective approach for correcting sun glint in high-resolution imagery [33,48].

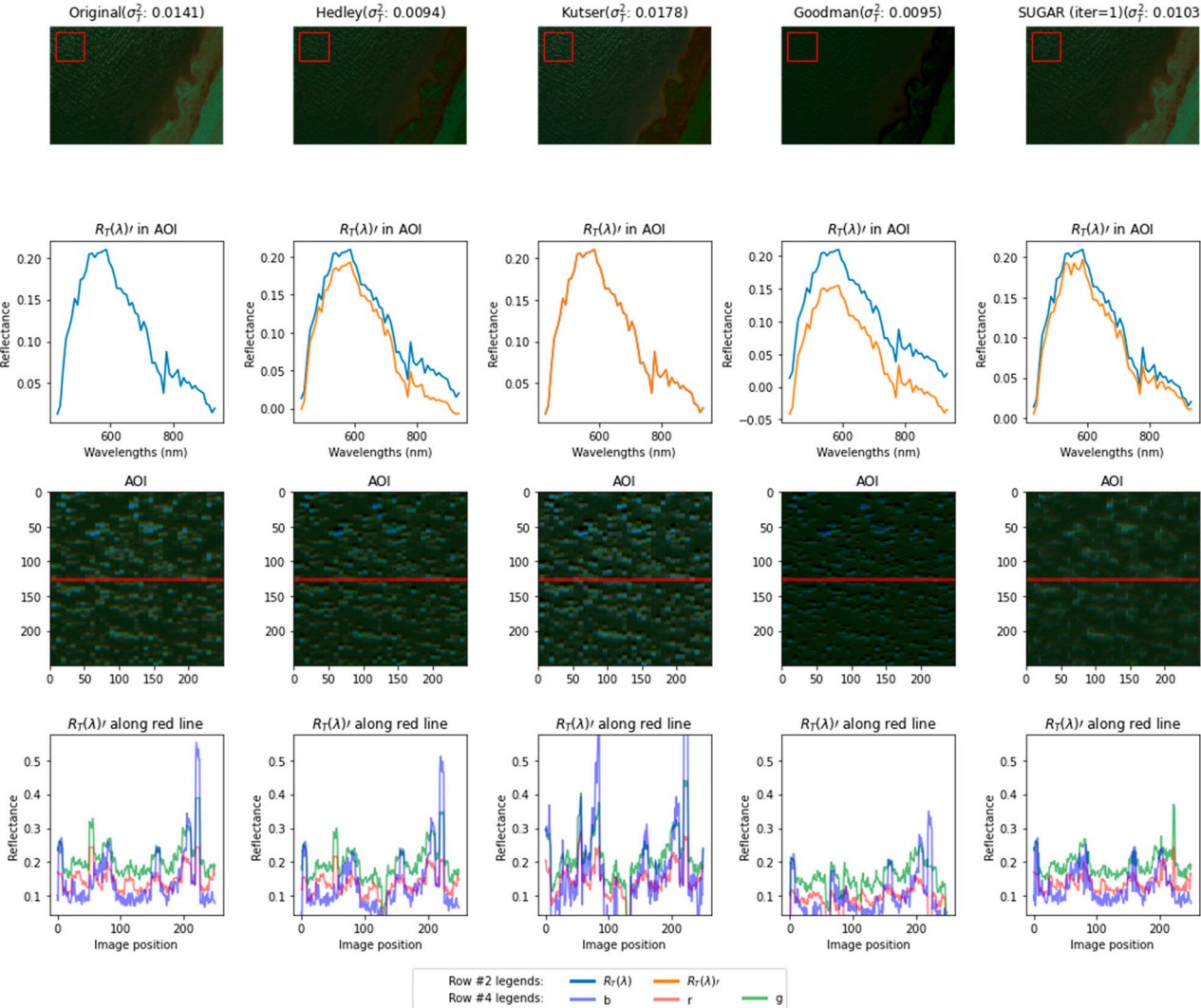

**Figure 11.** Comparison of traditional NIR-based sun glint correction and the sun glint correction employed in this study (Note: False composite mosaic created with blue = 476.37 nm, green = 568.56 nm, red = 669.37 nm).

In contrast to sun glint, where the presence of sun glint can at times dominate the at-sensor total reflectance, the presence of stripe artefacts in hyperspectral images can degrade the image quality and significantly reduce the signal-to-noise ratio. This was observed in Figure 12, where the stripe noises manifest as dark lines with varying thicknesses that occur intermittently throughout the image. Correcting for the stripe noises also significantly reduced the vignetting effects that occur around the boundary of the image, which improved the overall contrast of the images. This indicates that the method of de-striping using the calibrated white reference captured prior to every flight survey to obtain the prevailing intensity of the stripe noises was effective in the overall reduction of intrinsic noises in the sensor. While recent methods proposed by [49] involving the use of structure-guided unidirectional variation for hyperspectral imagers to perform the differentiation between stripe noises and edges/textures in the scene to detect the distribution of stripes effectively without the use of a white reference, the additional computation could increase image processing time significantly. The proposed de-striping method thus offers an efficient method of locating the stripe noises since a white reference that covers the entire field of view of the imager is already available and required for sensor calibration.

Before de-striping          After de-striping

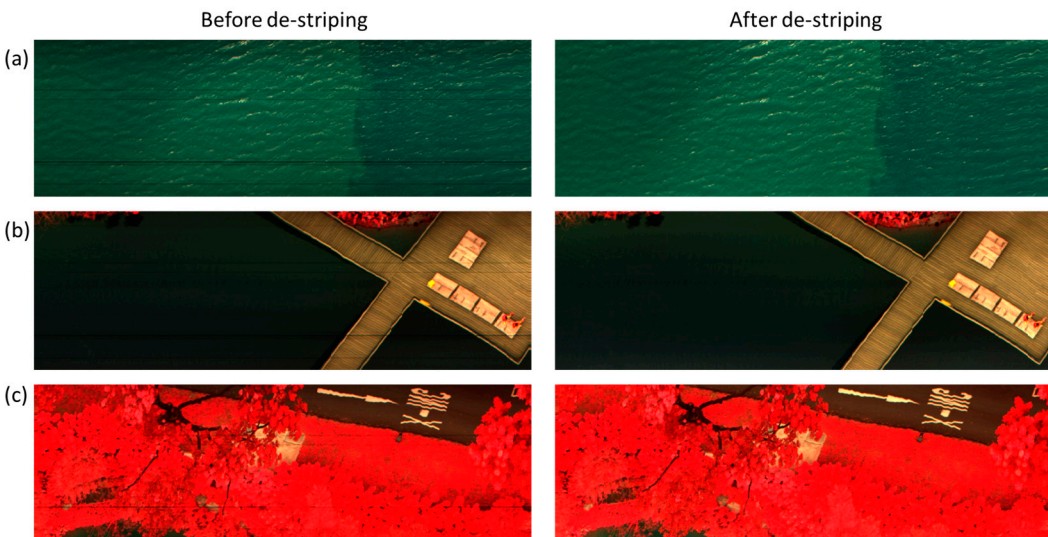

**Figure 12.** Removal of stripe noises under various scenarios: (**a**) coastal water in Singapore, (**b**) broadwalk in Nanyang Lake, and (**c**) road adjacent to Nanyang Lake (Note: False composite mosaic created with blue = 550 nm, green = 620 nm, red = 760.6 nm).

### 3.2. Evaluation of Individual Pre-Processing and Their Impact on Reflectance Spectrum

Figures 13 and 14 show the image mosaic and extracted spectral reflectances for each pre-processing step, with the sequence starting from the uncorrected reflectance images, followed by the time delay correction for image alignment, de-striping, radiometric correction, and finally, sun glint correction. Visually, it can be observed that the image quality improved significantly after the de-striping step, and the image brightness and reflectance were corrected based on the prevailing downwelling irradiance after conducting radiometric correction. There are, however, some brighter patches where radiometric correction was not sufficient, which could be attributed to the oversaturation of the spectroradiometer and, therefore, irradiance readings, leading to an under-correction in those circumstances. It is thus imperative that the calibration of the spectroradiometer should be conducted under the brightest illumination condition prior to the flight survey.

The resulting effect of de-striping and radiometric correction was also observed in the extracted spectral reflectances, which showed that the differences between the reflectances for high and low turbidity became more prominent (Figure 14). However, significant outliers can be visibly seen after the de-striping procedure, where the brightness may be inaccurately enhanced, which inflates the reflectances for some low turbidity measurements. Instability at the extreme wavelengths (e.g., lower end of visible spectrum < 500 nm and higher end of the NIR spectrum > 900 nm) was also observed after radiometric correction, where reflectances varied greatly. This is attributed to the poor signal-to-noise ratio of the sensor at these wavelengths, where the dark and white reference calibration curves lie very close to each other, and numerical instability may arise at these wavelengths (Figure 9).

It is also observed that the time delay correction to improve the image alignment between adjacent image swaths does not have a significant impact on the turbidity retrieval. The main reason may be due to the spatial distribution of the turbidity measurements while the sampling vessel traversed within the flight area, as shown in Figure 13. There were only two points where the sampling vessel traversed in and out of the turbidity plume, and only two image swaths captured these points that mark the sharp transition of the turbidity plume's edge. Therefore, any changes made to the image alignment only affect the turbidity measurements in these two image swaths. Nevertheless, the demarcation of the plume and non-plume regions became visually smoother after the time delay correction.

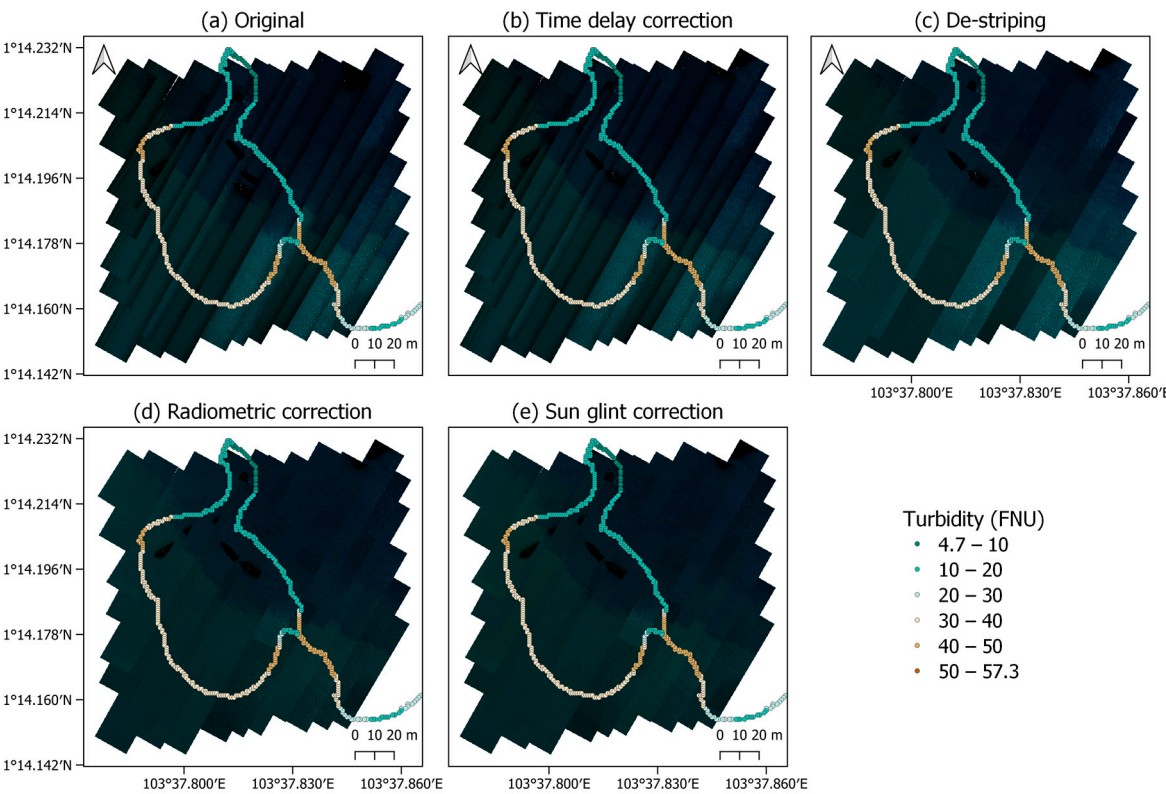

**Figure 13.** Reflectance image overlain with in-situ turbidity measurements at each pre-processing step (Note: vessels are masked for confidentiality).

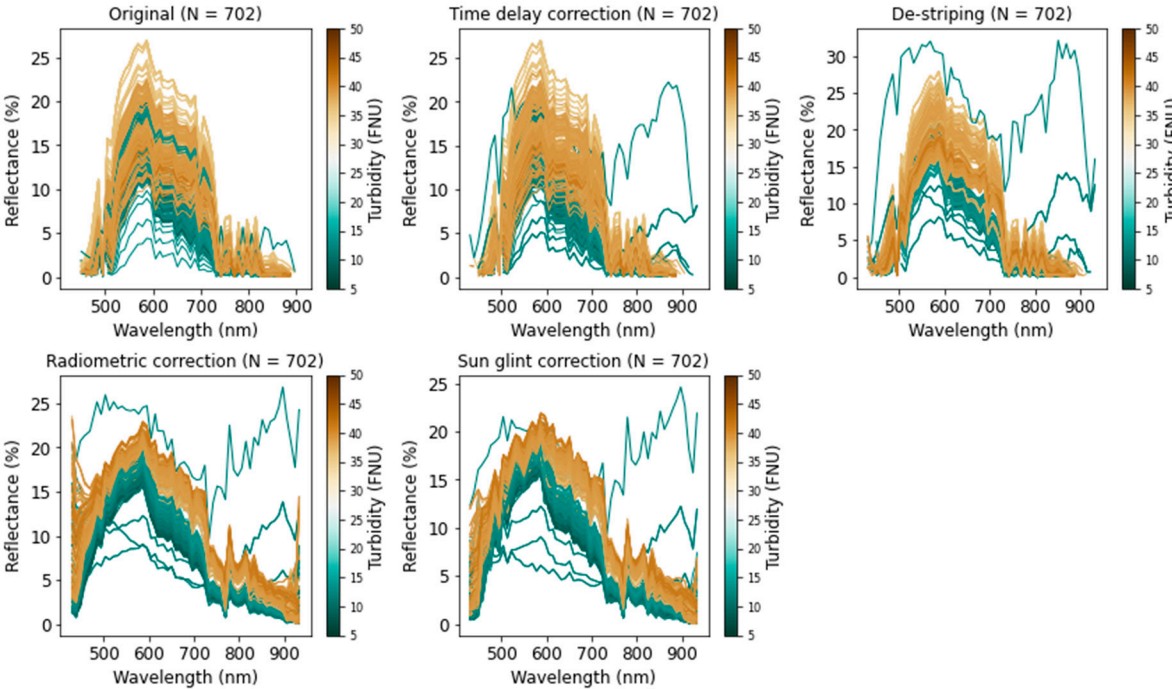

**Figure 14.** Extracted reflectance for each pre-processing step where the pre-processing sequence starts with masking, followed by the time delay correction for image alignment, de-striping, radiometric correction, and finally, sun glint correction.

### 3.3. Evaluation of Individual Pre-Processing and Their Impact on Turbidity Retrieval

Analysis was further conducted to analyse the effect of commonly reported bands for turbidity retrieval using Nechad et al.'s (2010) semi-analytical algorithm. Bands in the red to NIR wavelengths, such as 645 nm, 660 nm, 681 nm, 710 nm, and 860 nm were suggested by [4,39–41], and the corresponding closest wavelengths in the hyperspectral sensor (i.e., 641 nm, 660 nm, 678 nm, 715 nm, and 860 nm) were used to determine and assess the accuracy of turbidity retrieval. Table 2 reports the collated turbidity retrieval results using these wavelengths. The results show that the turbidity prediction improved steadily from 641 nm to 715 nm but dropped drastically at 860 nm.

**Table 2.** Turbidity retrieval results on various literature-reported red and NIR wavelengths.

| | 641 nm | 660 nm | 678 nm | 715 nm | 860 nm |
|---|---|---|---|---|---|
| (a) Original | RMSE = 9.894, MAPE = 0.492, $R^2$ = 0.458 | RMSE = 9.628, MAPE = 0.471, $R^2$ = 0.487 | RMSE = 8.594, MAPE = 0.368, $R^2$ = 0.591 | RMSE = 8.562, MAPE = 0.254, $R^2$ = 0.594 | RMSE = 30.145, MAPE = 0.943, $R^2$ = −4.534 |
| (b) Time delay correction | RMSE = 9.898, MAPE = 0.491, $R^2$ = 0.458 | RMSE = 9.612, MAPE = 0.469, $R^2$ = 0.489 | RMSE = 8.586, MAPE = 0.370, $R^2$ = 0.592 | RMSE = 8.568, MAPE = 0.251, $R^2$ = 0.594 | RMSE = 30.951, MAPE = 0.960, $R^2$ = −4.921 |
| (c) De-striping | RMSE = 8.641, MAPE = 0.427, $R^2$ = 0.587 | RMSE = 8.358, MAPE = 0.410, $R^2$ = 0.614 | RMSE = 7.279, MAPE = 0.340, $R^2$ = 0.707 | RMSE = 6.570, MAPE = 0.239, $R^2$ = 0.761 | RMSE = 30.230, MAPE = 0.972, $R^2$ = −4.330 |
| (d) Radiometric correction | RMSE = 5.943, MAPE = 0.275, $R^2$ = 0.805 | RMSE = 5.629, MAPE = 0.251, $R^2$ = 0.825 | RMSE = 5.175, MAPE = 0.229, $R^2$ = 0.852 | RMSE = 4.537, MAPE = 0.188, $R^2$ = 0.886 | RMSE = 21.338, MAPE = 0.701, $R^2$ = −1.519 |
| (e) Sun glint correction | RMSE = 5.856, MAPE = 0.274, $R^2$ = 0.810 | RMSE = 5.636, MAPE = 0.254, $R^2$ = 0.824 | RMSE = 4.885, MAPE = 0.211, $R^2$ = 0.868 | RMSE = 4.579, MAPE = 0.192, $R^2$ = 0.884 | RMSE = 21.861, MAPE = 0.750, $R^2$ = −1.644 |

Observed-predicted scatter plots for each pre-processing step are located in Appendix H.

From the table, band 715 nm achieved the best turbidity prediction results with the lowest RMSE and MAPE at 4.537 and 0.188, respectively, and the highest $R^2$ at 0.886. This is consistent with the finding of [39], which reported lower relative and absolute errors for wavelengths in the spectral range of 670–750 nm and hence recommended 681 nm for turbidity values in the range of 0.6–83 FNU (encompassing the range of turbidity values in this study from 8.41 to 41.56 FNU). Although the Nechad et al. (2010) algorithm was calibrated for MERIS bands and SPM measurements from the Southern North Sea, the applicability of the suggested band performed well for the in-situ turbidity measurements in this study after calibration of the coefficients $A_\tau$ and $C$.

Across all wavelengths, the greatest marginal improvement in turbidity prediction was attributed to radiometric correction, where RMSE decreased by at least 28% for wavelengths 641 nm to 715 nm (Table 2). This could indicate the importance of the radiometric correction for tropical regions where cloud cover is extensive and transient. Furthermore, it also indicates that the relative radiometric correction to convert DN directly into reflectance is adequate, provided that their calibration curve is well-established, as is the case in this study.

The drastic decrease in turbidity prediction at 860 nm seems to suggest higher instability at higher NIR wavelengths, which is similarly reflected in Figure 14, where the reflectance varied significantly beyond 750 nm, implying an intrinsic problem with the sensor calibration at higher NIR wavelengths rather than the calibration of the turbidity retrieval algorithm. As such, turbidity retrieval using wavelengths beyond 715 nm should not be recommended for Bayspec's hyperspectral sensor.

It is also observed that sun glint correction improved for bands 641 nm and 678 nm but showed no significant improvement for bands 660 nm and 715 nm. This is because the proposed sun glint correction corrects sun glint at each individual wavelength independently while existing sun glint correction algorithms such as [42–44] use the NIR band

as a surrogate for sun glint intensity, as well as the empirical relationship between NIR and other bands to correct for other bands. However, such a method has the unintended consequence of overcorrecting turbid regions in all bands where water-leaving NIR is non-negligible. On the other hand, the proposed sun glint correction in this study showed improvements in general, albeit not across all bands, while it did not significantly impact turbidity predictions.

From the generated turbidity map at bands 641 nm, 660 nm, 678 nm, and 715 nm (Figure 15), there was generally close correspondence between the in-situ turbidity measurements and the predicted turbidity, and the demarcation between low and high turbidity regions in the turbidity map similarly corresponded generally to the changes in the in-situ turbidity measurements. However, the predicted turbidity at 660 nm appeared to underestimate turbidity at regions with higher in-situ turbidity, while there was an overestimation of turbidity at these regions at 641 nm. The predicted turbidity at 678 nm and 715 nm lay in between and achieved slightly better prediction performance in Table 2. On the other hand, the predicted turbidity at 860 nm could not discern the turbidity plume, with poor results, as shown in Table 2. However, there is an image swath in particular which did not match the in-situ measurements and displayed an abrupt increase in turbidity, which also corresponds to the bright patch in Figure 13. This mismatch could be attributed to the under-correction in radiometric correction when the illumination conditions likely saturated the on-board spectroradiometer.

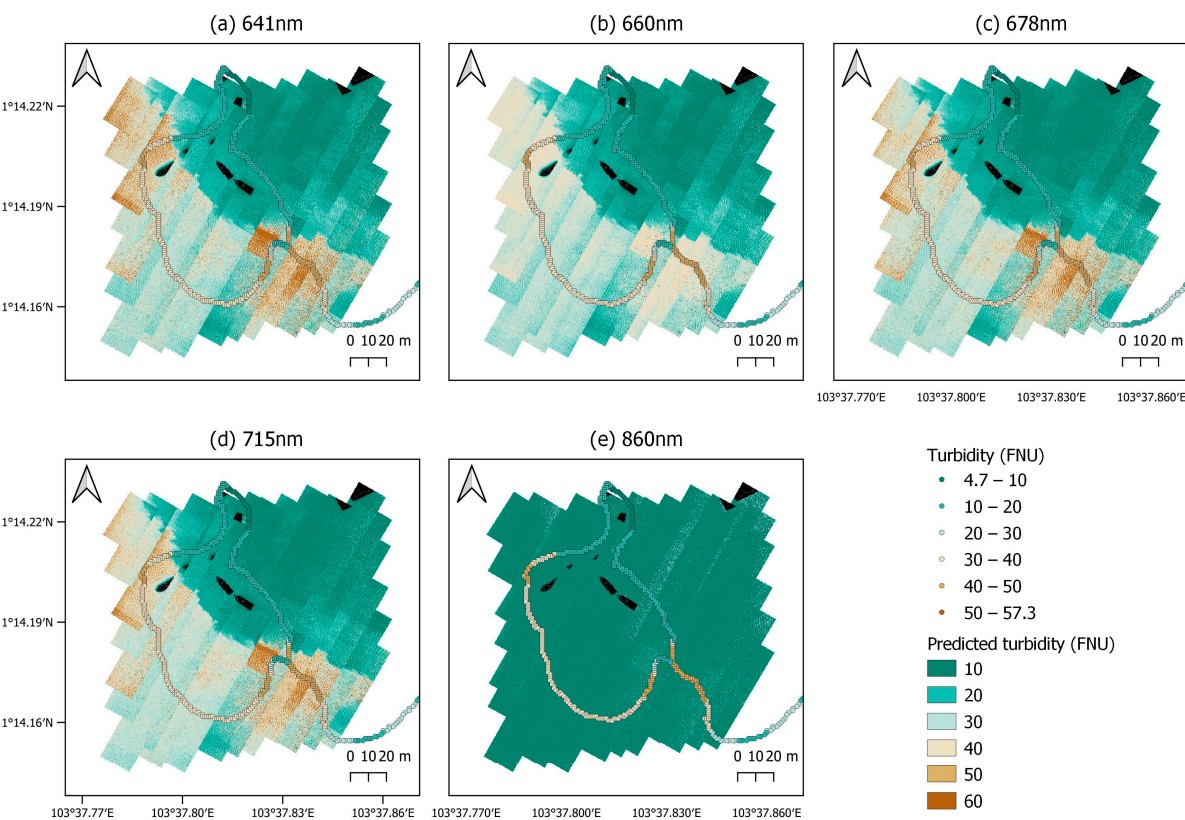

**Figure 15.** Predicted turbidity maps at bands (**a**) 641 nm, (**b**) 660 nm, (**c**) 678 nm, (**d**) 715 nm, and (**e**) 860 nm after time delay correction, de-striping, radiometric correction, and sun glint correction have been applied on the reflectance imagery.

## 4. Discussion

### 4.1. Retrieval of Turbidity

Turbidity retrieval from UAV imagery with machine learning (ML) models appears to show promise. In another UAV study by [50], where various ML models were evaluated

for the retrieval of suspended solids (SS), SS prediction in ranges from $R^2 = 0.91$ to 0.99, which performed significantly better than some studies using semi-analytical algorithms, e.g., [4,21]. With the development of CoastalWQL, its application, together with ML models, was used to extensively monitor turbidity plumes associated with land reclamation activities with an $R^2 = 0.75$ (see [5]). However, training of ML models typically requires a large dataset, and the choice of different ML models requires fine-tuning of various hyperparameters.

On the other hand, turbidity retrieval with semi-analytical algorithms such as [39] can be largely generalized for various coastal and estuarine waters at the red to NIR wavelengths as it is underpinned by the inherent optical property (IOP) of turbidity. For instance, the results obtained in this study using Nechad et al.'s (2010) [39] semi-analytical algorithm at 715 nm showed close correspondence with the results obtained in a similar study that estimated dredge-induced turbidity plume using the same semi-analytical algorithm at 660 nm, where the study achieved a RMSE of 3.39 with a UAV-borne MSI (Parrot Sequoia) for turbidity values ranging from 3.3 to 72.4 FNU [4]. Another study using the same semi-analytical algorithm for turbidity retrieval at 668 nm obtained an RMSE of 10.13 FNU with the MicaSense Dual Camera System [21].

However, such semi-analytical algorithms for turbidity retrieval can also be sensitive to particle size, particle density, and refractive index since they influence the nature of the backscattering [40]. As such, recalibration of the coefficients may be necessary as the particle size distribution of the study site may not necessarily be representative of other turbid coastal sites. The coefficients derived for this study thus vary slightly from the recommended coefficients in [39]. The re-calibrated coefficients for the in-situ turbidity at 715 nm were 137.85 and 0.2516 for $A_\tau$ and $C$, respectively, while the suggested coefficients by [39] were 708.16 and 1.17 for $A_\tau$ and $C$, respectively. The recalibration of the coefficients is required since $A_\tau$ and $C$ relate to IOPs such as absorption by particle and non-algal particles and particulate backscatter, which depends on the morphology and characteristics of the particles, as well as turbidity concentration. Recalibration of $A_\tau$ and $C$ coefficients were similarly conducted by [21] where the derived coefficients are 366.14 and 0.1956 for $A_\tau$ and $C$ at 668 nm.

### 4.2. Existing Limitations and Challenges in Image Mosaicking and Alignment

Some UAV-borne sensor manufacturers, such as Bayspec, adopt a more modular system where other problems may arise, such as the time delay between the GNSS and the imager, where the misalignment due to time delay between the two modules can result in significant misalignment. Further research is thus required to automatically estimate this time delay. Also, it should be acknowledged that some features are still required to ascertain the optimal time delay, where a feature is captured in at least two adjacent overlapping swaths, and the same time delay correction can be applied across all the swaths. In extreme cases where the water surface is perfectly homogenous, the time delay correction is limited, and users will have to rely purely on the GNSS's accuracy. As such, wherever possible, using an RTK/PPK system on the UAV is highly recommended to increase the georeferencing accuracy.

Geometric distortion is a prevalent problem in the mosaicking of push-broom hyperspectral imagery, as several factors such as GPS coordinates, timestamps, IMU offsets, lens characteristics, and altitude offset parameters can influence the mosaicking procedure [51]. Such geometric distortion is even more challenging to correct in the absence of distinctive features in the scene. As such, stitching images along the swath is the most efficient method [31]. Additionally, in the OCI-F push-broom sensor, each image band's dimension is only $1024 \times 20$, and thus, the resolution of the overlap ratio is $\frac{100\%}{20 \; pixels} = 5\%$ per pixel, which does not provide a lot of degree of freedom for fine-tuning the overlap ratio in adjacent images, which could easily introduce geometric distortions in the mosaicked image.

The above geometric distortion could be more pronounced if the images are taken when the FPS is too low or too high, which could lead to some scenes being cut off entirely in the mosaicked image or the elongation of the image, respectively. Additionally, the presence of sudden strong winds can cause the UAV system to drift slightly off from its course as the IMU in the UAV systems may not respond fast enough for course correction. As such, image mosaicking with minimal geometrical distortions is best achieved when conducting image stitching along a straight line [14]. As such, during flight planning, the "lawn-mowing" pattern is recommended to reduce geometrical distortions. Furthermore, wind drag and turbulence can vary the UAV's flight height and course and, therefore, introduce geometrical distortions in the mosaicked image [28]. Based on past field experiences in conducting UAV flight surveys in the coastal environment, both scenarios could occur—where the FPS of the push-broom HSI decreased significantly in the midst of a flight survey due to rapidly approaching dark clouds that reduced the prevailing illumination condition, as well as strong wind speeds (>8 m/s), which at times could render the flight imagery completely unusable. Under such challenging scenarios in the coastal region, some scenes were completely cut off in the mosaicked image, and radiometric correction would be severely limited due to the poor signal-to-noise ratio in the HSI and spectroradiometer.

### 4.3. Comparisons with Existing Methods/Software

CoastalWQL workflow framework was evaluated against recent techniques and existing software and summarised in Table 3. Orthomosaicking and direct-georeferencing techniques recently introduced by VITO's MapEO water [21] and MosaicSeadron [22] utilised direct-georeferencing on each snapshot RGB/multispectral image. An orthomosaic of the scene is generated by merging the rasters thereafter. However, this technique differs from the procedure for orthomosaicking of push-broom imagery in several ways, and they should be clearly differentiated as follows.

Firstly, MosaicSeadron's [22] method is underpinned by the availability of the georeferencing parameters for each image. In MicaSense's multispectral imagery, each image is tagged with the geographical coordinates and altitude in the metadata, where an image and its geographical coordinates are captured simultaneously every 3–5 s [21]. In push-broom imagery; however, the average frames per second (FPS) during a flight is 50 FPS (i.e., one capture per 20 ms), while the geographical coordinates are captured at one-second intervals, and thus the temporal resolution of the captured images and geographical coordinates differs greatly compared to that of the MicaSense's multispectral imager. As such, only some push-broom images that are captured simultaneously with the GPS measurements have geographical information. This implies that the mosaicking in between the measured geographical coordinates has to be conducted along the flight swath prior to direct georeferencing, where the mosaicking parameters are determined by the UAV flight parameters, as similarly suggested by [31]. Despite differences in the techniques, the direct georeferencing error rates achieved by both MosaicSeadron [22] and CoastalWQL are comparable, at standard deviations of 2.51 m and 2.69 m, respectively (Table 3).

Secondly, the nature of the image obtained from different sensors plays a significant role in the processing of the images. Each band image dimension from a push-broom sensor is significantly smaller than that of a snapshot imagery (Table 3 and Appendix A), as light from the observed strip of terrain is only allowed to enter through a small slit in the entrance port [24,52]. The number of images to cover a given survey area is, therefore, significantly higher due to the narrow dimension of the image and a required higher FPS (the average number of images taken by the Bayspec push-broom sensor to cover a ground coverage of around 10,000 m$^2$ is 30,000–40,000, compared to around 600 images with the MicaSense multispectral sensor in [21]). As such, direct-georeferencing of mosaicked push-broom images is more computationally efficient for a large number of images compared to individual push-broom images. The very narrow image dimension of an individual push-broom image also means that fewer overlapped features can be captured, which is

required for SfM techniques unless a secondary snapshot imager is used and co-registered simultaneously with the push-broom imager.

In summary, the techniques for mosaicking and conducting direct-georeferencing of featureless water body images from the snapshot and push-broom sensors are vastly different due to the inherent differences in their sensors and data output. The differences are detailed in the software solutions, as shown in (Table 3). Commercial software for mosaicking snapshot UAV imagery, such as PIX4D Mapper and Agisoft Metashape, accepts common image formats such as JPEG, TIFF, BMP, etc. On the other hand, the processing of push-broom hyperspectral imagery typically requires specialised software (that is usually provided upon purchase with the sensor) to process images that are binary files and are organised in various schemas, depending on the manufacturer of the sensor (e.g., Band Interleaved by Line (BIL), Band Interleaved by Pixel (BIP), and Band Sequential (BSQ)). As such, mosaicking of push-broom imagery is often incompatible with the aforementioned commercial software. In the case of the push-broom imagery in this study, the processing of the imagery is compatible with Bayspec's CubeCreator and CubeStitcher, although a failure in the mosaicking of the images frequently occurs over water bodies due to the lack of distinctive features, which prevents further processing with other software such as Agisoft Metashape (see Appendix I).

**Table 3.** Comparison of CoastalWQL with existing UAV software/processing workflow for water quality mapping.

| | Bayspec's Cube Creator | VITO's MapEO Water [21] | Agisoft Metashape, PIX4D Mapper | MosaicSeadron [22] | CoastalWQL |
|---|---|---|---|---|---|
| **(a) Software implementation** | | | | | |
| Software environment | Windows | Cloud platform | Windows, Mac | Python notebook | Windows (GUI) via python script, python notebook |
| Sensor | Bayspec's OCI-F | Micasense RedEdge-MX, DJI P4 multispectral | Micasense RedEdge-MX, Micasense Dual Camera System, etc. | DJI H20T sensor, DJI Mavic 2 Enterprise Advanced (M2EA) thermal sensor, Micasense Dual Camera System | Bayspec's OCI-F |
| Data input | Push broom hyperspectral (binary file) | Snapshot multispectral (.tiff) | Snapshot multispectral/RGB (.tiff, PNG, JPEG, BMP, etc.) | Snapshot multispectral (.tiff) | Push broom hyperspectral (binary file) |
| Image dimensions | 1024 × 20 | 1280 × 960 (MicaSense), 5472 × 3648 (DJI P4) | Various e.g.,1280 × 960 (MicaSense), etc. | 1280 × 960 (MicaSense), 640 × 512 (M2EA and H20T) | 1024 × 20 |
| Open-source | No | No | No | Yes | Yes |
| **(b) Pre-processing workflow** | | | | | |
| Image mosaicking | Feature-based image mosaicking, e.g., SfM | Merging of rasters | Feature-based image mosaicking, e.g., SfM, image alignment | Merging of rasters | Image mosaicking along flight swath ([31]) (independent of scene's features) |
| Georeferencing | Georeferencing via GPS coordinates, flight parameters | Direct-georeferencing | Georeferencing via image registration (e.g., GCPs) | Direct-georeferencing | Direct-georeferencing |

**Table 3.** *Cont.*

| | Bayspec's Cube Creator | VITO's MapEO Water [21] | Agisoft Metashape, PIX4D Mapper | MosaicSeadron [22] | CoastalWQL |
|---|---|---|---|---|---|
| Error rate in direct-georeferencing | NA (stitching failure over some scenes) (Appendix I) | Not published | NA | $\sigma = 2.51$ m at GSD ~ 0.5 m/px | $\sigma = 2.69$ m at GSD ~ 0.2 m/px |
| Conversion to reflectance product | Empirical relative radiometric correction | Radiometric conversion to convert DN into radiance, and into reflectance using [53] | Radiometric conversion to convert DN into radiance, and into reflectance (by providing reflectance panel) | NA | Empirical relative radiometric correction |
| Masking/ classification | User-defined classification threshold for classification | Masking of non-water pixels | Additional user-defined processing | NA | Classification and masking of land/caisson and vessels |
| Image alignment | NA | NA | Image alignment via GCPs | NA | (optional) Time delay image alignment with real-time visualisation |
| Correction of intrinsic noises | NA | Correction of lens vignetting effects via MicaSense's image processing library | Correction of lens vignetting effects (MicaSense imagery) | NA | De-striping |
| Atmospheric correction | NA | iCOR4Drones | NA | NA | NA |
| Sun glint correction | NA | NA | NA | NA | Modified SUGAR algorithm |
| Water quality product | NA | Turbidity, suspended sediments, chlorophyll | NA | NA | Turbidity |

Notes: Comparison between the image mosaic from Bayspec's Cube Creator and CoastalWQL is found in Appendix I (Note: NA represents that the method is not available and/or has to be implemented outside the software/processing workflow).

### 4.4. Limitations of CoastalWQL and Future Directions

CoastalWQL so far has only been validated with in-situ measurements for turbidity, and validation with other water quality parameters such as CDOM and chlorophyll-a was not carried out in our study due to its key objective to determine turbidity attributed to dredging and sediment-dumping operations. Further validation is thus required in the future for investigating different various water body objectives. While CoastalWQL is developed for use for Bayspec's OCI-F hyperspectral push-broom sensor, it is envisioned that the open-source nature of CoastalWQL can encourage the adoption and adaptation of the pre-processing workflow for users who similarly experience the problems of image mosaicking over featureless water body and image misalignment due to the time delay between the GNSS and the imager.

Due to the plethora of commercial UAV-borne sensors, challenges still remain in streamlining processing workflow, and many of this workflow are inevitably sensor and platform-specific. There is thus a need for greater open access to UAV data and workflows such that the development of software and water retrieval algorithms for UAV imagery

could be more streamlined for higher applicability to various sources of UAV imagery. As such, there is a need for commercial UAV-borne sensor manufacturers and users to follow the Findability, Accessibility, Interoperability and Reusability (FAIR) principles to improve reproducibility [54,55]. For example, the Multiscale Observation Networks for Optical Monitoring of Coastal Waters, Lakes and Estuaries (MONOCLE) ecosystem follows the FAIR principles for water quality monitoring through various in situ sensors and provides data access for streamlined monitoring by agencies and research organisations.

Other challenges in sharing UAV data include potential infringement of personal data and privacy issues [54], the massive data storage required on a cloud platform, maintenance of assets and standardization of metadata. Such practices are standardized for satellite products, but are largely lacking for UAV products. However, platforms such as the Open Aerial Map are one of the first few open services to make headway in this promising direction by providing access to openly licensed imagery. Other organisations such as MicaSense have open-sourced their code, which demonstrates how various metadata is stored and accessed in the images and, therefore, provides greater customizability for the consumers. However, such open-source frameworks for push-broom hyperspectral data are still relatively rare so far, as hyperspectral systems produced by various OEMs typically produce their own proprietary software to manage and process various output data. With further transparency and sharing of frameworks, the adoption of UAV imagery for monitoring/mapping applications can be further enhanced in the future.

## 5. Conclusions

In this study, CoastalWQL is developed as an automated workflow for UAV-hyperspectral water quality monitoring to address the problem of image mosaicking over largely homogenous surfaces, especially under restrictive conditions in the coastal environment where image alignment via GCPs is not possible on the water surface and land features are not always available. In these circumstances, various previous studies typically employed their own pre-processing routine, which can be an arduous process requiring significant time and resources. To promote the common adoption of UAV-hyperspectral monitoring in the coastal environment, CoastalWQL has been made open-source to provide a complete pipeline for the reflectance products, with essential pre-processing procedures such as radiometric correction, masking of non-water objects and sun glint correction for water quality monitoring applications.

To validate the applicability of the developed workflow, coastal turbidity monitoring was conducted in Singapore using a UAV-borne push-broom hyperspectral sensor, and CoastalWQL was applied for the aforementioned pre-processing procedures. Turbidity retrieval using a semi-empirical model was conducted at each step of the pre-processing workflow to evaluate the effectiveness of each pre-processing procedure, and the retrieved turbidity was validated against in-situ turbidity measurements. Turbidity retrieval was found to be the most optimal at 715 nm, and turbidity prediction improved by 46.5%, with RMSE reduced significantly from 8.562 FNU to 4.579 FNU and $R^2$ improving from 0.594 to 0.884 after applying the essential pre-processing workflow. It was observed that de-striping of the hyperspectral imagery and radiometric correction provided the largest marginal improvement to turbidity retrieval and are essential pre-processing steps.

CoastalWQL aims to facilitate and address similar issues experienced by the community involved in UAV-based hyperspectral water quality monitoring. It is also envisioned that other open-source frameworks for processing push-broom hyperspectral imagery in monitoring water quality via UAVs and push-broom HSIs can subsequently be created in a similar manner to improve accessibility for the public, industry and researchers.

**Author Contributions:** H.Y.P.: Conceptualisation, Methodology, Software, Formal analysis, Visualisation, Writing—Original draft. H.T.K.: Methodology, Formal analysis, Writing—Reviewing and Editing. W.L.: Writing—Reviewing and Editing. E.K.: Writing—Reviewing and Editing, Funding acquisition. A.W.-K.L.: Supervision, Resources, Project administration, Funding acquisition, Writing—Reviewing and Editing. All authors have read and agreed to the published version of the manuscript.

**Funding:** This research was funded by the Singapore Maritime Institute (SMI) under the research project "UAV-based Remote Sensing of Turbidity in Coastal Waters", grant number SMI-2020-MA-02.

**Data Availability Statement:** CoastalWQL (https://github.com/pakhuiying/CoastalWQL) (accessed on 13 February 2024).

**Acknowledgments:** We would like to extend our gratitude to Li Min, Dawn Pang, and Trinh Ha Linh for the procurement of equipment and data acquisition and to the Environmental Process Modelling Centre (EPMC) for providing the equipment and facilities to conduct the relevant work. The authors would also like to acknowledge the generous assistance in project coordination and data collection provided by the Maritime and Port Authority of Singapore (MPA), DHI Water and Environment (S) Pte Ltd., and Surbana Jurong Pte Ltd. (SJ), under the research project funded by the Singapore Maritime Institute (SMI), "UAV-based Remote Sensing of Turbidity in Coastal Waters", grant number SMI-2020-MA-02. The first author would also like to thank Public Utilities of Singapore (PUB)—Singapore's National Water Agency, for granting the scholarship for the PhD study. This research is supported by the National Research Foundation, Singapore, and PUB, Singapore's National Water Agency under its RIE2025 Urban Solutions and Sustainability (USS) (Water) Centre of Excellence (CoE) Programme, awarded to Nanyang Environment & Water Research Institute (NEWRI), Nanyang Technological University, Singapore (NTU).

**Disclaimer:** Any opinions, findings and conclusions or recommendations expressed in this material are those of the author(s) and do not reflect the views of National Research Foundation, Singapore and PUB, Singapore's National Water Agency.

**Conflicts of Interest:** The authors declare no conflicts of interest.

## Appendix A

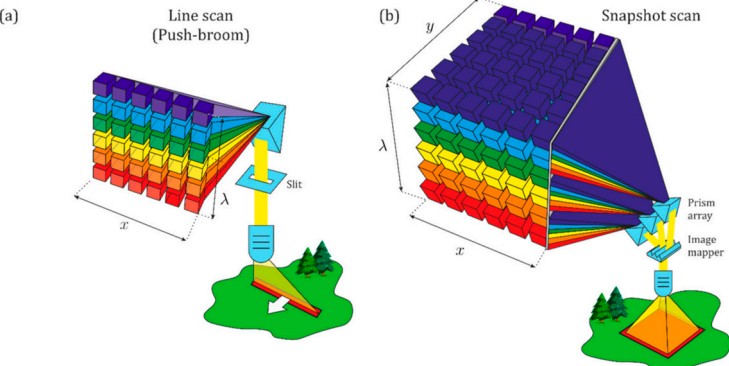

**Figure A1.** Differences in imaging principles between push-broom and snapshot imagery (**a**) narrow slit of the image and scene captured in push-broom imager, (**b**) larger extent of the image and scene captured via snapshot imager (Source: [24], which is adapted from [23]).

## Appendix B

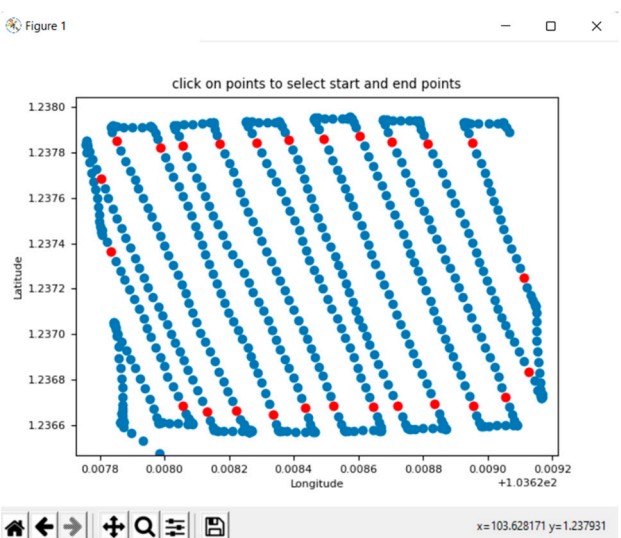

**Figure A2.** GPS selection tool in CoastalWQL to facilitate the image mosaicking between any selected two GPS coordinates (Note: Red points are any pair of user-selected GPS points along the swath).

## Appendix C

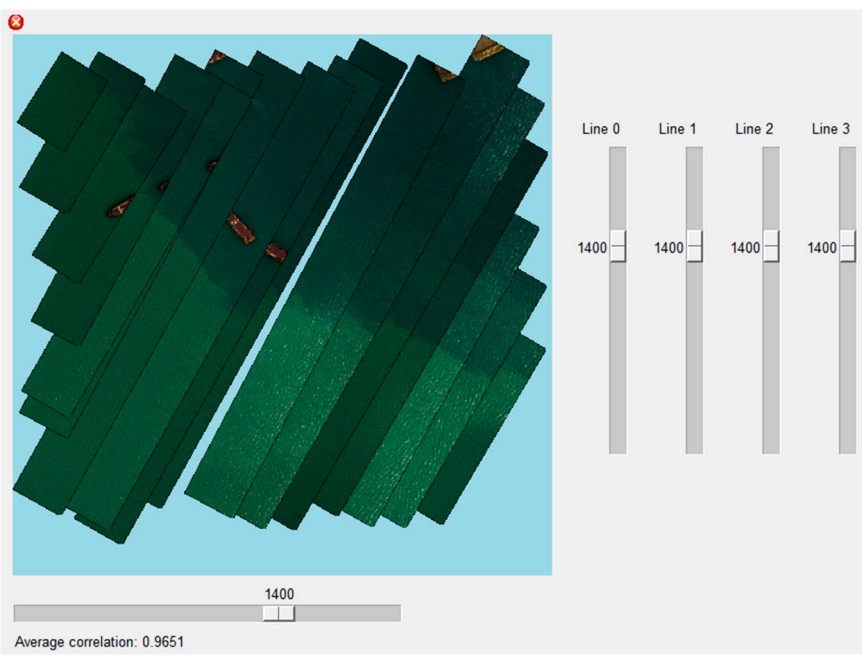

**Figure A3.** Image alignment tool in CoastalWQL for fine-tuning of the time delay correction.

## Appendix D

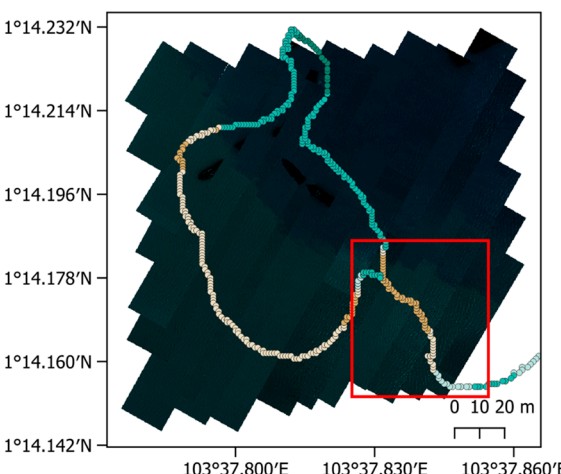

**Figure A4.** Sampling points highlighted in the red box overlapped spatially due to the sampling vessel traversing the same area at least once. (Note: sampling points in the red box were thus removed).

## Appendix E

**Table A1.** Inputs and outputs of *CoastalWQL*.

| (i) Inputs | Data |
|---|---|
| Image folder | Path to the folder directory containing the folder of the raw hyperspectral images and the folder of GPS file, where the raw hyperspectral images are in .raw format, and GPS file is in .csv format. |
| Flight region in GUI | A GUI window where user can specify the range of flight regions to conduct mosaicking |
| Height | User-specified height (in metres) at which the drone operates and when imaging is conducted. GPS information entails the altitude information but not the height information, where altitude = height + altitude of surface topography |
| Spectrometer folder | A series of text files (.txt), each containing the absolute irradiance information ($\mu W/cm^2/nm$) at wavelengths covering the entire spectral range of the hyperspectral camera |
| Water quality data | A csv file containing information of the in-situ water quality measurements. It should contain columns with the measurements of the water quality concentration, and two other columns with its corresponding latitude and longitude information |
| Trained model | An exported trained model in .JSON or .model (for XGBoost models) format that contains trained model parameters |
| (ii) Outputs | |
| False composite image | The user is given the flexibility to choose three wavelengths to represent the RGB channels. Output image has a .tif format |
| Masked image | An image that has been masked to conceal vessels at the study site for confidentiality |
| Geo-registered/georeferenced image | An image that has been transformed from the image coordinate space to the georeferenced coordinate space (.tif) |
| Extracted spectral information | If the water quality data is provided, spectral information is extracted based on the supplied coordinates |

## Appendix F

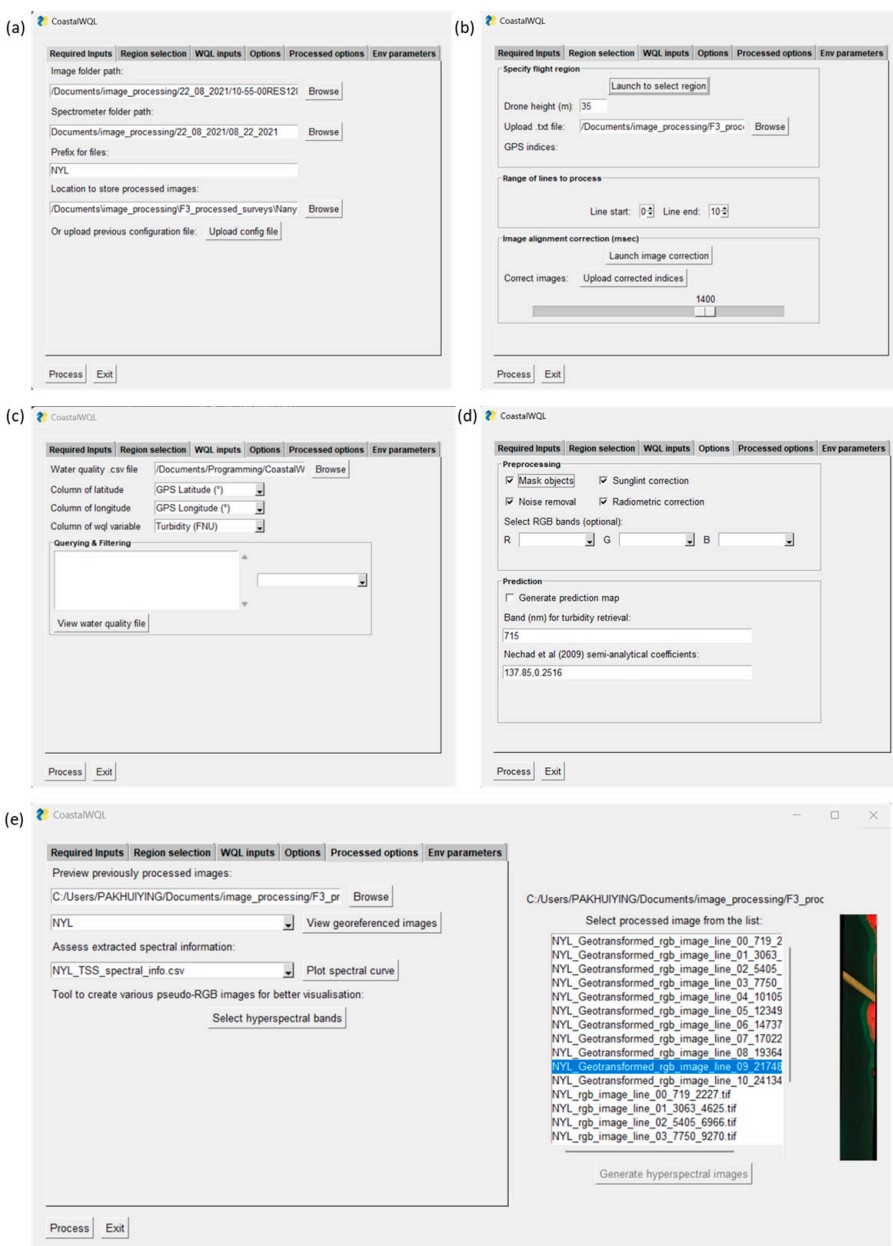

**Figure A5.** CoastalWQL functionalities (**a**) Inputs (**b**) Selection of GPS coordinates along image swath for mosaicking (**c**) Input of in situ water quality data (**d**) additional pre-processing options (**e**) viewing processed images and extracted spectrum.

## Appendix G

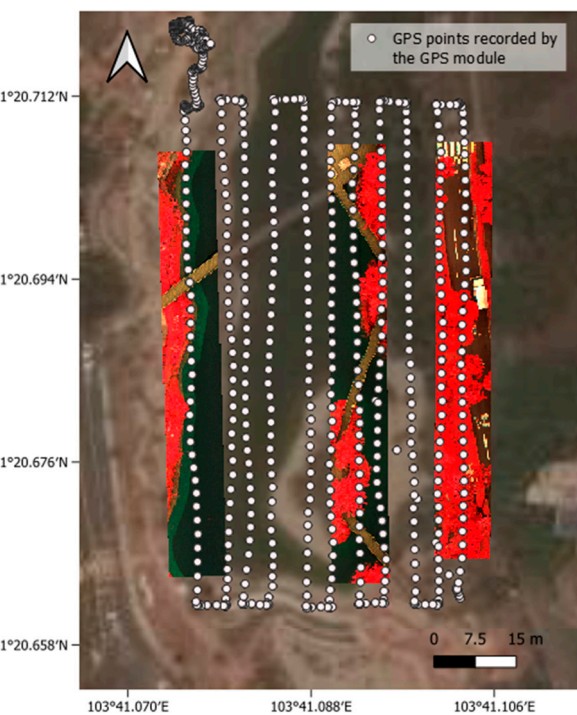

**Figure A6.** GPS points along the imaging swath (Note: Non-overlapping individual flight swaths are shown).

## Appendix H

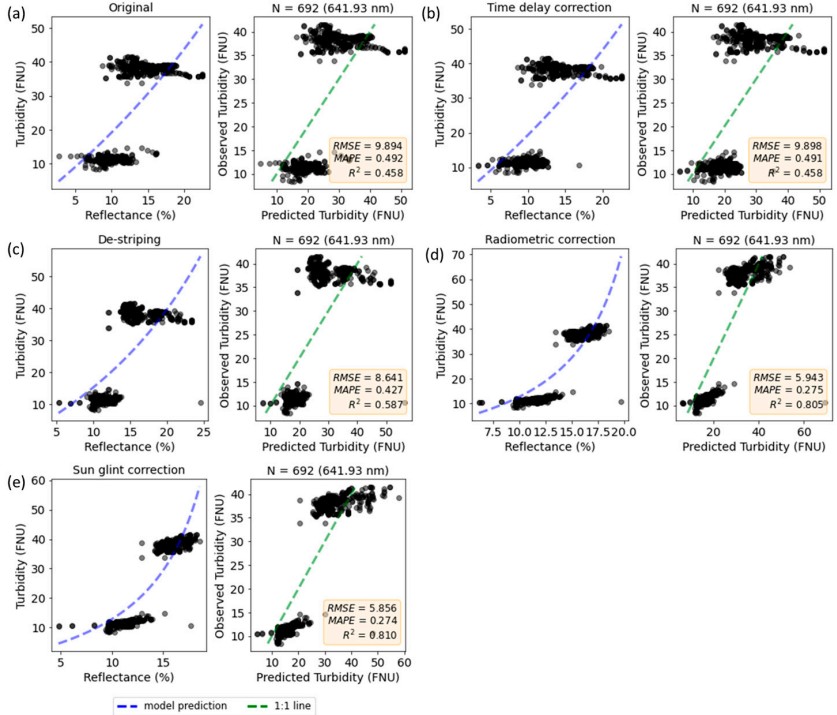

**Figure A7.** Observed versus predicted turbidity for each pre-processing step at band 641 nm (**a**) original, (**b**) time delay correction for image alignment, (**c**) de-striping, (**d**) radiometric correction, and (**e**) sun glint correction (Note: left panel of sub-figure is a plot of turbidity concentration against reflectance, right panel of sub-figure is a plot of observed versus predicted turbidity concentration).

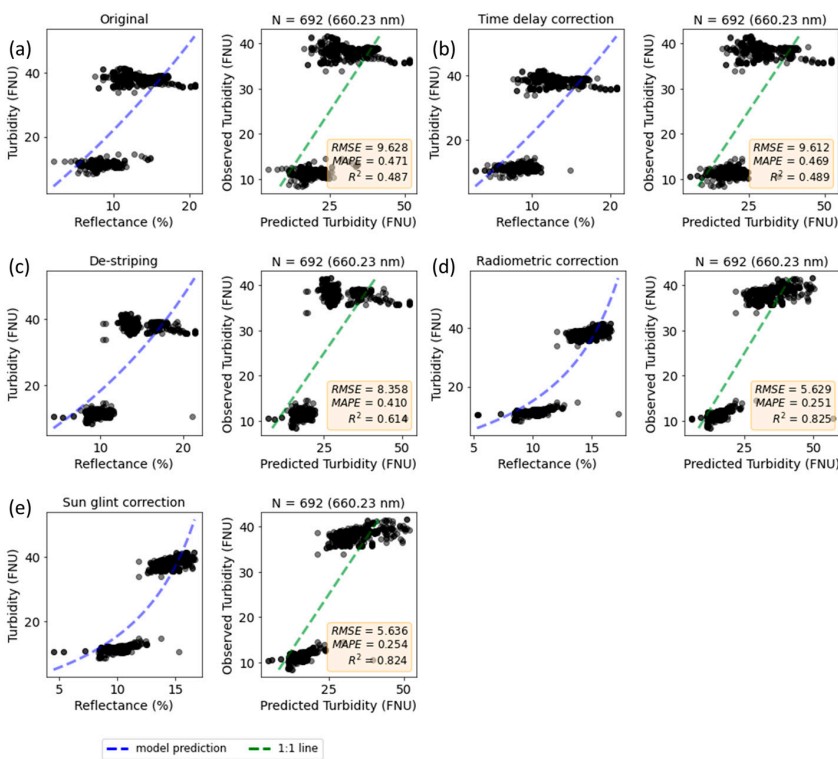

**Figure A8.** Observed versus predicted turbidity for each pre-processing step at band 660 nm (**a**) original, (**b**) time delay correction for image alignment, (**c**) de-striping, (**d**) radiometric correction, and (**e**) sun glint correction (Note: left panel of sub-figure is a plot of turbidity concentration against reflectance, right panel of sub-figure is a plot of observed versus predicted turbidity concentration).

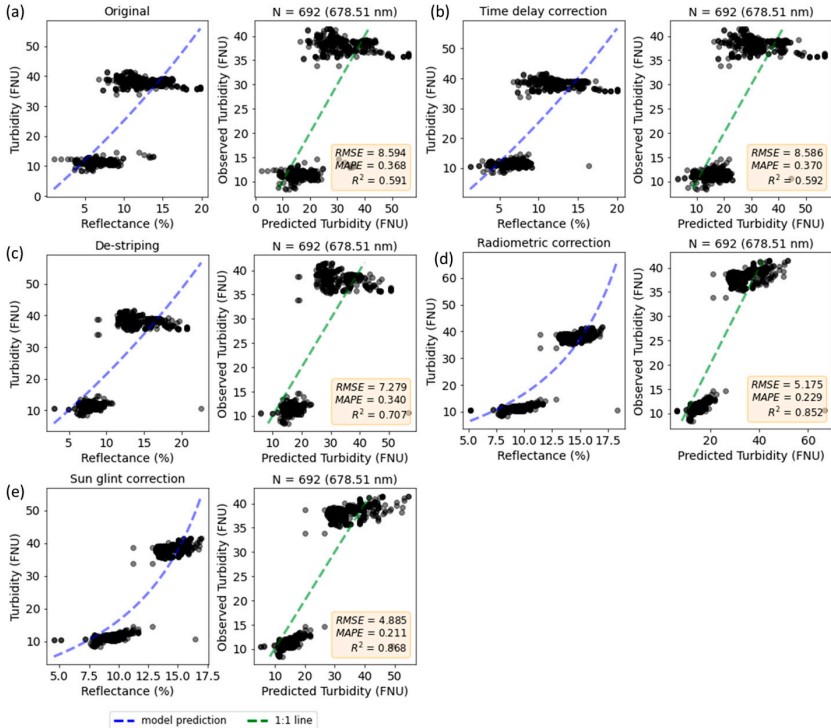

**Figure A9.** Observed versus predicted turbidity for each pre-processing step at band 678 nm (**a**) original, (**b**) time delay correction for image alignment, (**c**) de-striping, (**d**) radiometric correction, and (**e**) sun glint correction (Note: left panel of sub-figure is a plot of turbidity concentration against reflectance, right panel of sub-figure is a plot of observed versus predicted turbidity concentration).

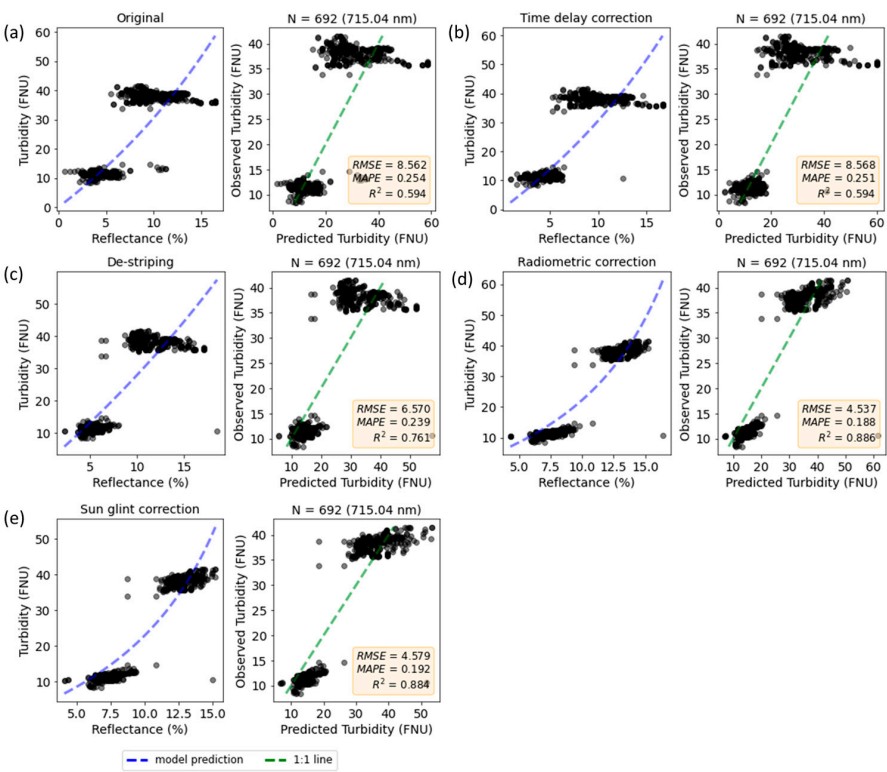

**Figure A10.** Observed versus predicted turbidity for each pre-processing step at band 715 nm (**a**) original, (**b**) time delay correction for image alignment, (**c**) de-striping, (**d**) radiometric correction, and (**e**) sun glint correction (Note: left panel of sub-figure is a plot of turbidity concentration against reflectance, right panel of sub-figure is a plot of observed versus predicted turbidity concentration).

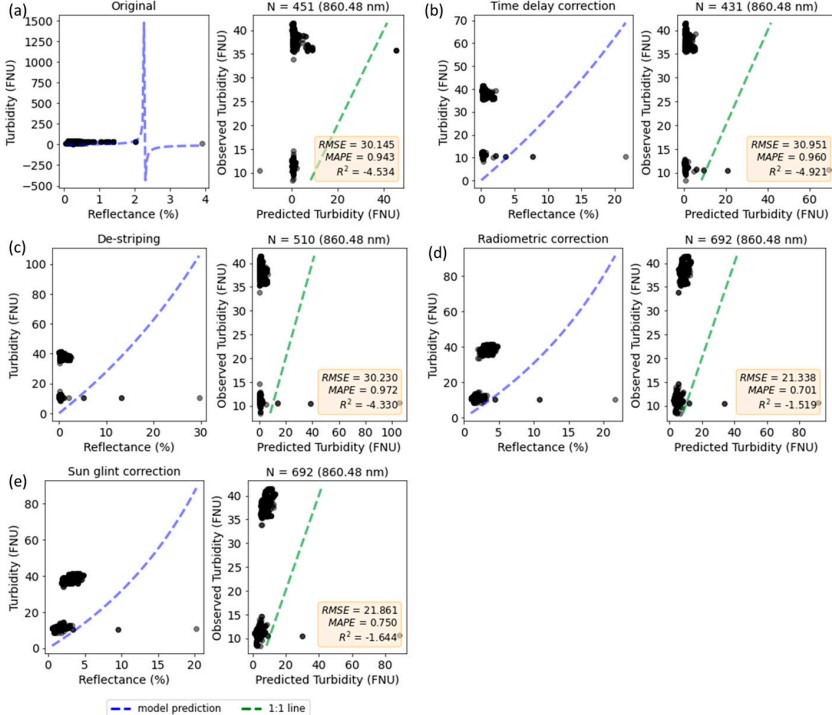

**Figure A11.** Observed versus predicted turbidity for each pre-processing step at band 860 nm (**a**) original, (**b**) time delay correction for image alignment, (**c**) de-striping, (**d**) radiometric correction, and (**e**) sun glint correction (Note: left panel of sub-figure is a plot of turbidity concentration against reflectance, right panel of sub-figure is a plot of observed versus predicted turbidity concentration).

# Appendix I

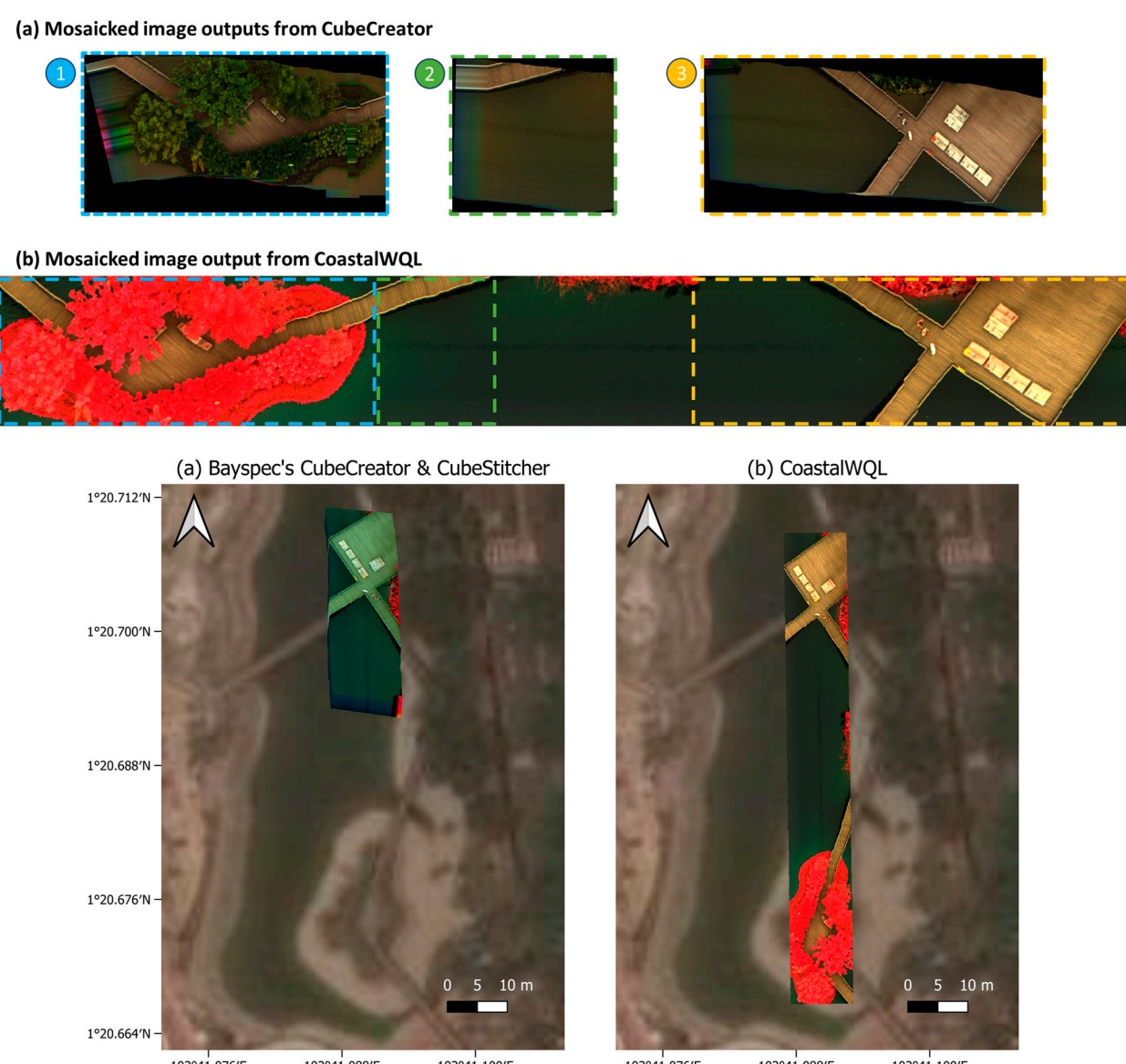

**Figure A12.** Mosaicked image output from Bayspec's CubeCreator and CubeStitcher, in comparison with the output from CoastalWQL (Notes: the mosaicked image outputs from CubeCreator were not successfully stitched in CubeStitcher and Agisoft Metashape to form an image line due to lack of tie points and distortions at the image edges).

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
