# Peer review of "CoastalWQL: An Open-Source Tool for Drone-Based Mapping of Coastal Turbidity Using Push Broom Hyperspectral Imagery"

_remotesensing, doi:10.3390/rs16040708_

Round 1
Reviewer 1 Report
Comments and Suggestions for Authors
see attached

Comments on the Quality of English LanguageYou can express your high English proficiency and clear expression in the article。
Reviewer 2 Report
Comments and Suggestions for Authors
The paper focuses on addressing two significant challenges in UAV-based hyperspectral water quality monitoring: mosaicking featureless coastal water surfaces and correcting systematic image misalignment due to GPS time delay. This study develops and presents an open-source toolkit, CoastalWQL, which simplifies the workflow for pre-processing and processing push-broom hyperspectral data for water quality monitoring. This will increase accessibility and encourage wider adoption of the technology. The paper validates the effectiveness of CoastalWQL using a UAV-based turbidity monitoring study in Singapore. Quantitative improvements in turbidity retrieval after applying the pre-processing workflow demonstrate its efficacy.
Your research addresses an important issue, and your proposed algorithm shows promise.
The title effectively summarizes the research and target audience.
The abstract clearly outlines the problem, solution, and findings.
The methodology section provides sufficient detail about the workflow and CoastalWQL functionalities.
The results section effectively presents the validation study and improvement in turbidity retrieval.
The conclusions reiterate the significance of CoastalWQL and its potential for broader application.
However, there are several areas that could be improved to enhance the clarity and rigor of your work. Here are the specific suggestions for revision
While open-source tools are mentioned, a detailed comparison of CoastalWQL's functionalities and performance with existing alternatives would strengthen the paper.
The paper could elaborate on the limitations of CoastalWQL, such as its applicability to different water bodies and types of sensors. Discussing potential future improvements and research directions would be valuable.
This article introduces the research of multiple scholars and explains their research methods. However, the literature section can emphasize the shortcomings of various methods, which is enough to shift to newly proposed methods. It can also incorporate various Review papers on the topic. It is better to add the following references to enrich the work:
Zhong, Y., Wang, X., Xu, Y., Jia, T., Cui, S., Wei, L., ... & Zhang, L. (2017, July). MINI-UAV borne hyperspectral remote sensing: A review. In 2017 IEEE International Geoscience and Remote Sensing Symposium (IGARSS) (pp. 5908-5911). IEEE.
Parsons, M., Bratanov, D., Gaston, K. J., & Gonzalez, F. (2018). UAVs, hyperspectral remote sensing, and machine learning revolutionizing reef monitoring. Sensors, 18(7), 2026.
Lu, Q., Si, W., Wei, L., Li, Z., Xia, Z., Ye, S., & Xia, Y. (2021). Retrieval of water quality from UAV-borne hyperspectral imagery: A comparative study of machine learning algorithms. Remote Sensing, 13(19), 3928.
Mishra, V., Avtar, R., Prathiba, A. P., Mishra, P. K., Tiwari, A., Sharma, S. K., ... & Jain, K. (2023). Uncrewed Aerial Systems in Water Resource Management and Monitoring: A Review of Sensors, Applications, Software, and Issues. Advances in Civil Engineering, 2023.
Authors are also advised to avoid the term "unmanned" in expansion of UAV. Recently there has been emphasis on the usage of inclusive and gender-neutral terms. You can use 'uncrewed' instead.
line 570-571: "Distinctive features were identified as GCPs in the Google satellite imagery’s tiles". Justify using it since there are positional inaccuracies in these images.
Reviewer 3 Report
Comments and Suggestions for Authors
I had an opportunity to review a paper “CoastalWQL: An open-source tool for drone-based mapping of coastal water quality using push broom hyperspectral imagery”, which seemed to be interesting.
However, the paper is neither review, article nor methodical type.
In overall it is far too long, first 5 pages which are introduction, are typical for the review papers. In my opinion a large portion of this section is unnecessary for the purpose of this article.
There is no scientific aim or hypothesis, which including non-scientific rest of the paper disqualify it as a research article.
Discussion section is basically an introduction all over again. Please provide a discussion according to the subject of the MS. There is no validation of the method as a water quality tool, just a technical overview. Results are not discussed and there is no reference to the literature on the subject.
Please consider shortening the manuscript, and modify it significantly to the proper research paper.
Round 2
Reviewer 2 Report
Comments and Suggestions for Authors
Authors have addressed all the queries.
Reviewer 3 Report
Comments and Suggestions for Authors
The paper has been revised according to the reviewer's suggestions. It now meets the criteria for publication.
